# The ER calcium channel Csg2 integrates sphingolipid metabolism with autophagy

Shiyan Liu[1,9], Mutian Chen[2,3,4,9], Yichang Wang[5,9], Yuqing Lei[6], Ting Huang[1], Yabin Zhang[1], Sin Man Lam ®[7,8], Huihui Li ®[6] ✉, Shiqian Qi[5] ✉, Jia Geng ®[2,3,4] ✉ & Kefeng Lu ®[1] ✉

Sphingolipids are ubiquitous components of membranes and function as bioactive lipid signaling molecules. Here, through genetic screening and lipidomics analyses, we find that the endoplasmic reticulum (ER) calcium channel Csg2 integrates sphingolipid metabolism with autophagy by regulating ER calcium homeostasis in the yeast *Saccharomyces cerevisiae*. Csg2 functions as a calcium release channel and maintains calcium homeostasis in the ER, which enables normal functioning of the essential sphingolipid synthase Aur1. Under starvation conditions, deletion of Csg2 causes increases in calcium levels in the ER and then disturbs Aur1 stability, leading to accumulation of the bioactive sphingolipid phytosphingosine, which specifically and completely blocks autophagy and induces loss of starvation resistance in cells. Our findings indicate that calcium homeostasis in the ER mediated by the channel Csg2 translates sphingolipid metabolism into autophagy regulation, further supporting the role of the ER as a signaling hub for calcium homeostasis, sphingolipid metabolism and autophagy.

Calcium is a ubiquitous extracellular and intracellular ion that influences almost every aspect of cell functions[1,2]. The concentration of calcium is maintained at very low levels in the cytosol (0.1 μM), and a steep calcium gradient exists across the plasma membrane (extracellular space calcium at 1 mM) and the membranes of intracellular calcium reservoirs (endosome calcium at 2 μM, acidic lysosome calcium at 0.5 mM, endoplasmic reticulum calcium at 0.5 mM, Golgi apparatus calcium at 0.1 mM, mitochondria calcium at 0.1 μM and secretory vesicle calcium at 0.1 mM)[2–4]. Calcium is the only ion that can boast such characteristics[5]. To enable prompt responses to various internal or external signals, the cellular cytosolic concentration of calcium is increased by influx through pumps in cell membranes or by release from intracellular calcium reservoirs through ion channels[6–9]. Cytosolic calcium binds to and activates calcium-modulated proteins (calmodulins) that transduce signals into appropriate downstream outputs through calmodulin-binding proteins (calcineurin), calmodulin-dependent protein kinases, and histone deacetylases[10,11]. Calcium-permeable ion channels contribute to rapid changes in cytosolic calcium concentrations by enabling calcium influx into or out of the environment, and by determining the intracellular pathways for calcium uptake into or release from cellular organelles such as lysosomes/vacuoles and the ER[7,12,13].

Thus, calcium channels indirectly regulate a variety of calcium-related cellular processes, including cell proliferation, apoptosis, and

[1]Department of Neurosurgery, State Key Laboratory of Biotherapy, West China Hospital, Sichuan University, Chengdu 610041, China. [2]Department of Laboratory Medicine, State Key Laboratory of Biotherapy, West China Hospital, Sichuan University, Chengdu 610041, China. [3]Department of Laboratory Medicine, State Key Laboratory of Biotherapy, Med-X Center for Manufacturing, West China Hospital, Sichuan University, Chengdu 610041, China. [4]Tianfu Jincheng Laboratory, City of Future Medicine, Chengdu 641400, China. [5]Department of Urology, State Key Laboratory of Biotherapy, West China Hospital, Sichuan University, Chengdu 610041, China. [6]Department of Pathology, West China Second University Hospital, Sichuan University, Chengdu 610041, China. [7]State Key Laboratory of Molecular Developmental Biology, Institute of Genetics and Developmental Biology, Chinese Academy of Sciences, Beijing 100101, China. [8]LipidALL Technologies Company Limited, Changzhou 213022, China. [9]These authors contributed equally: Shiyan Liu, Mutian Chen, Yichang Wang. ✉e-mail: lihuihui@scu.edu.cn; qishiqian@scu.edu.cn; geng.jia@scu.edu.cn; lukf@scu.edu.cn

autophagy[5,8]. For example, increasing calcium activates calcineurin to dephosphorylate TORC2 and activate PKC and RasGRP to initiate the MAPK cascade, eventually activating genes that regulate proliferation[14]. Under stress conditions, the mitochondrial matrix is rapidly overloaded with calcium, which is the first step of apoptosis induction[15,16]. The roles of calcium in autophagy are controversial. Studies have shown that calcium promotes autophagy by inhibiting the activity of mTOR through CAMKK2/CaMKKb (calcium/calmodulin dependent protein kinase kinase 2) and AMPK[17–20], and other studies have shown that an increase in intracellular calcium negatively regulates autophagy by promoting the binding of $Ca^{2+}$-CALM/calmodulin to PIK3C3/VPS34 and subsequent mTOR activation[21,22]. Compared to the roles of calcium in the cytosol, the consequences of abnormal calcium accumulation inside reservoir organelles caused by dysfunctional calcium channels are less well understood[1,3,5,6,8,23]. Studies have shown that disruption of calcium channels in organelles such as the ER and mitochondria is correlated with abnormal insulin reactions and metabolic alterations in the liver[24–26]. Obesity-induced dysfunctional SERCA, an ER calcium channel, causes disrupted calcium homeostasis in the ER and impaired protein folding capacity[27]. The importance of such a mechanism in the development of fatty liver and diabetes is further supported by studies showing that SERCA overexpression enhances the protein folding function of the ER, leading to upregulated glycemic control[28]. Excessive calcium accumulation has been found to be associated with mitochondrial dysfunction[29,30]. The expression of the calcium channel IP3R was increased significantly in the mitochondria-associated ER membranes (MAMs) of liver cells from different models of obesity[31]. Such alteration caused increased calcium flux from the ER to mitochondria, and this calcium overload resulted in increased mitochondrial reactive oxygen species (ROS) production and decreased oxidative phosphorylation[29,30]. Consistently, suppression of IP3R restored mitochondrial function and significantly improved glucose tolerance in obese model mice[31].

Autophagy is a conserved intracellular process by which the cell degrades and recycles cytosolic constituents[32–37]. Generally, autophagy is found to play a predominant role in cell survival, especially under stress conditions such as starvation[38–41]. Autophagy functions to maintain normal cellular homeostasis by eliminating various cargoes, including misfolded proteins, protein aggregates, redundant or damaged organelles, and invading pathogens, which helps cells mobilize/recycle various energy sources and thus provides necessary nutrients and removes potentially dangerous elements[42–45]. The autophagy process has been found to be essentially involved in a number of physiological processes, such as embryo development, cell differentiation, oocyte fertilization, innate and adaptive immunity, cell death, and life-span extension[32,42,46–48]. Therefore, dysfunctional autophagy (mainly with decreased autophagic activity) is related to various diseases, including infectious, metabolic, cardiovascular, pulmonary, and neurodegenerative diseases and cancers[49–51]. The roles and mechanisms of calcium in regulating the autophagy process are rather complicated and controversial[5,8,13,52]. While many reports suggest an inhibitory effect of calcium on autophagy, there are also contrasting reports showing a stimulatory action of calcium on autophagy[8,13,53–59].

Considering that calcium levels in the cytosol and reservoir organelles are regulated by calcium channels, discovering the connections between certain calcium channels and autophagy can help clarify the mechanisms by which calcium regulates autophagy.

Sphingolipids represent a major class of conserved eukaryotic lipids that are not only ubiquitous components of the membranes of eukaryotic cells but also function as signaling bioactive lipid molecules[60–62]. Bioactive lipids are tightly involved in cellular regulatory circuits, which distinguish them from lipids with structural and energetic functions[60,63]. Among various sphingolipids, bioactive sphingolipids, such as ceramide (Cer), sphingosine (Sph), and Sph

1-phosphate (S1P), have been implicated in many key cellular processes, such as cell growth, proliferation, death, endocytosis, nutrient acquisition, and protein trafficking[64,65]. Dysregulation of bioactive sphingolipid levels has been reported in many diseases, such as cancers, diabetes, and neurodegenerative diseases[60,63,66]. Mutations in sphingolipid metabolism enzymes have also been linked with pathologies, including lysosomal storage disorders[67,68]. The upstream steps in the biosynthesis of sphingolipids are conserved in eukaryotic organisms from yeasts to metazoans[69]. The elements and mechanisms that regulate sphingolipid metabolism and maintain sphingolipid homeostasis in eukaryotic cells remain largely unexplored.

Here, we established a previously undescribed link that integrates calcium accumulation inside the ER with dysfunctional sphingolipid metabolism and blocked autophagy in the yeast *Saccharomyces cerevisiae*. We show that the ER calcium channel Csg2 is essential for restraining the calcium concentration inside the ER because it directs calcium flux out of the ER. Under starvation conditions, deletion of Csg2 causes an increase in ER calcium, which disturbs sphingolipid synthesis and causes the accumulation of a particular bioactive sphingolipid, phytosphingosine (PHS). Increased PHS levels completely block autophagy. Consistently, starvation resistance conferred by autophagy is inhibited by calcium accumulation in the ER and disrupts sphingolipid anabolism. Therefore, our findings suggest that information about the ER calcium status of the cell is fed into the regulation of sphingolipid synthesis and autophagy by the calcium channel Csg2.

## Results

### Csg2 is essential for starvation resistance and autophagy

To analyze the involvement of calcium channels in autophagy regulation, we screened *Saccharomyces cerevisiae* yeast cells with deletion of various calcium channels (confirmed and potential) (Fig. 1a). We searched for potential autophagy factors by observing cell viability after amino acid starvation. Autophagy confers starvation resistance under nutrient limitation conditions[39,70]. The results showed that deletion of Csg2, a putative ER-resident calcium channel protein[71–73], specifically induced the loss of starvation resistance in yeast cells subjected to amino acid starvation (SD-N) (Fig. 1b). Consistently, deletion of Csg2 but no other calcium channels completely blocked autophagy, as shown by GFP-Atg8 processing assays[74] in which the cleaved GFP moiety indicated autophagic degradation cargoes in vacuoles/lysosomes (Fig. 1c). Therefore, we focused on Csg2. Autophagic degradation of GFP-Atg8 upon starvation was monitored by detecting the cleaved GFP moieties, and the results showed that in *csg2Δ* cells the autophagic degradation of GFP-Atg8 was blocked as in *atg1Δ* cells, indicating that Csg2 is essential for autophagy (Fig. 1d). Additional autophagic substrates and assays were used to confirm the essential role of Csg2 in autophagy using the known essential factor Atg1[75–78] as a positive control. The degradation of GFP-50Q, another substrate of autophagy[79–81], was shown to be blocked in Csg2-deleted yeast cells (Fig. 1e). Ape1, a classic autophagy substrate[82–84], was not transported into vacuoles in Csg2-deleted yeast cells under starvation conditions (Fig. 1f). Quantitative alkaline phosphatase assays[74,85] were carried out, and the results showed that Csg2 deletion blocked the autophagic transfer of the substrate Pho8Δ60 into vacuoles to form active alkaline phosphatase (Fig. 1g). In addition to amino acid starvation, glucose starvation-induced autophagy was also blocked in Csg2-deleted yeast cells (Fig. S1a). Exogenously expressed Csg2 completely restored autophagy in Csg2-deficient yeast cells, as demonstrated by cell viability assays and GFP processing assays (Fig. S1b–d).

Next, we determined which steps of autophagy were regulated by Csg2 and whether Csg2 affects vesicle pathways other than autophagy. The blockage of autophagosome formation by Atg1 deletion resulted in a diffuse cytosolic distribution of Atg8, whereas deletion of the SNARE subunit Vam3 or GTPase Ypt7 caused accumulation of GFP-Atg8 puncta[86,87], indicating the failure of autophagosome fusion with

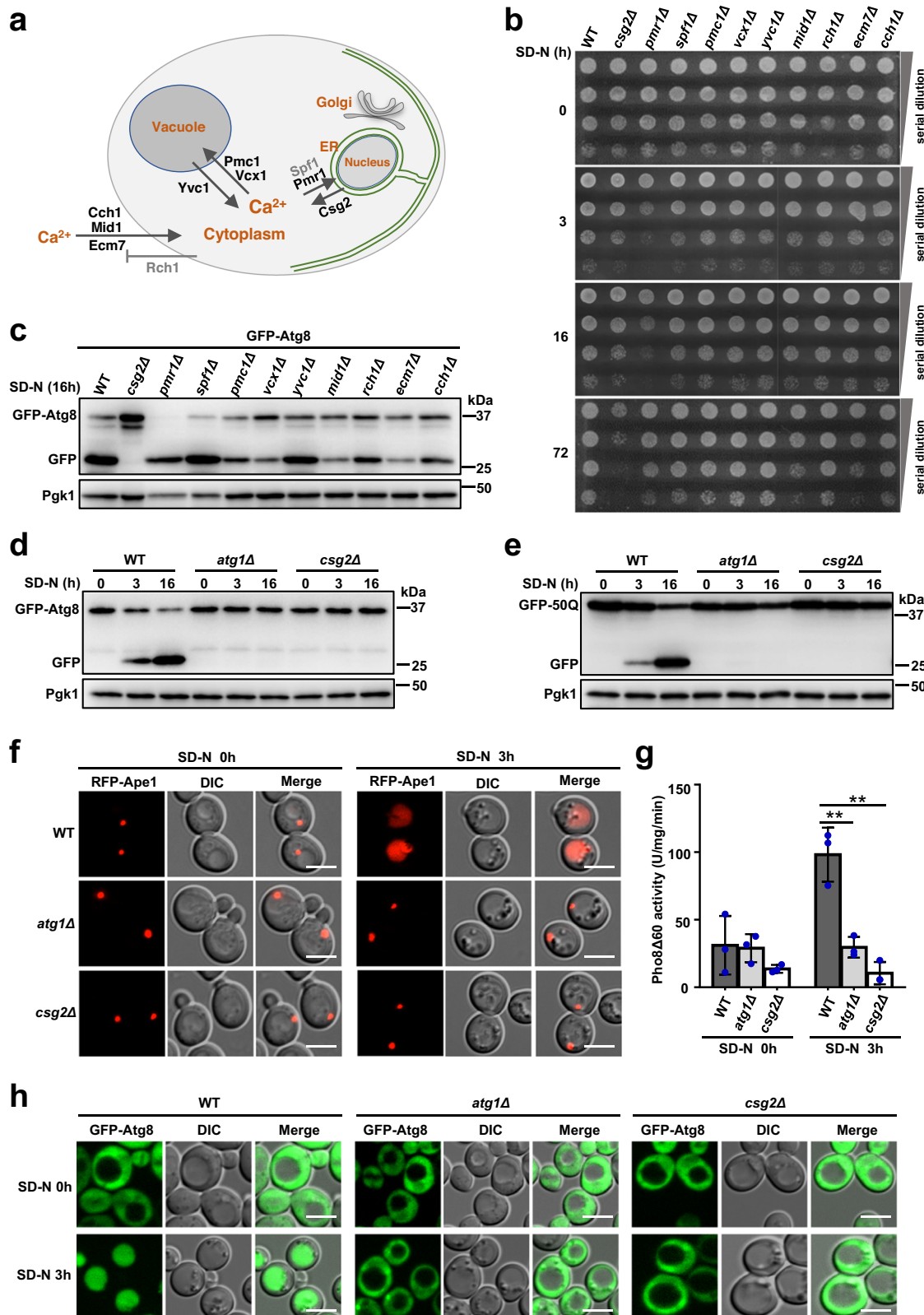

vacuoles (Fig. 1h and Fig. S1e). Upon Csg2 deletion, a diffuse cytosolic distribution of Atg8 was observed (Fig. 1h), suggesting that Csg2 is involved in regulating the formation of autophagosomes. Furthermore, we found that Csg2 was not involved in the vacuolar transport of the endocytic substrate GFP-Sna3 (Fig. S1f, g) or in the Golgi sorting and trafficking of carboxypeptidase Y/CPY into vacuoles (Fig. S1h, i), indicating that Csg2 was specifically involved in autophagy. Under nutrient-rich conditions, the Ape1 processing was not affected by Csg2 deletion (Fig. S1j), suggesting that the function of Csg2 in regulating autophagy was specific to starvation conditions. Deletion of Pmr1, a high-affinity calcium ATPase that transports calcium into the ER, caused upregulation of autophagy (Fig. S1k), suggesting that transport directions of calcium (into the ER by Pmr1 and out of the ER by Csg2) exerted opposite effects on autophagy.

**Fig. 1 | Csg2 is identified essential for autophagy. a** Scheme of calcium pumps and channels in yeast cells. **b** WT and calcium pumps/channels-deleted yeast cells including csg2Δ cells, pmr1Δ cells, spf1Δ cells, pmc1Δ cells, vcx1Δ cells, yvc1Δ cells, mid1Δ cells, rch1Δ cells, ecm7Δ cells, cch1Δ cells were starved for different times in SD-N medium and then detected for cell viability by serial dilution spotting and growth on YPD plates. The experiment was repeated independently for three times with similar results. **c** N-terminal GFP-tagged Atg8 with the ADH promoter was transferred into WT and calcium pumps/channels-deleted yeast cells to test its autophagic degradation after 16 h of starvation in SD-N medium. The blots were probed with anti-GFP antibody and Pgk1 was used as a loading control. The samples derive from the same experiment and gels were processed in parallel. **d** Validation of degradation of autophagic substrate GFP-Atg8 in WT cells, atg1Δ cells, and csg2Δ cells following starvation in SD-N medium was performed at the indicated time points. The blots were probed with anti-GFP antibody and Pgk1 was used as a loading control. The samples derive from the same experiment and gels were processed in parallel. **e** Validation of Csg2-dependent degradation of autophagic substrate GFP-50Q (50 poly-glutamine) as in (**d**). **f** N-terminal RFP-tagged Ape1 was transferred into vacuoles in WT cells after starvation, but not in atg1Δ cells and csg2Δ cells detected by fluorescence assays. The experiment was repeated independently for three times with similar results. Representative images were shown. Scale bars: 5 μm. **g** Detection of Csg2-dependent transportation of the autophagic substrate Pho8Δ60 by ALP assays. Indicated yeast cells were subject to starvation in SD-N medium. After transfer into vacuoles by autophagy, Pho8Δ60 activities were measured. Two-tailed t-test, $n = 3$ independent experiments. $p$ values: 0.00518, 0.00217. **$p < 0.01$. Error bars, mean ± SD. **h** Autophagic marker GFP-Atg8 under control of endogenous ATG8 gene promoter was investigated by fluorescence examination in WT cells, atg1Δ cells and csg2Δ cells before and after starvation. The experiment was repeated independently for three times with similar results. Representative images were shown. Scale bars: 5 μm.

From these lines of evidence, we conclude that Csg2, among various calcium channels, specifically plays an essential role in autophagy by regulating the formation of autophagosomes.

## Csg2 is an ER-resident calcium channel mediating calcium release

Csg2 has been suggested to be required for the synthesis of complex sphingolipids[88–91] and as a potential channel for calcium[71–73]. Sequence analysis and protein structure prediction revealed that Csg2 mainly exhibited 10 transmembrane (TM) domains and a potential EF-hand motif for calcium binding (Fig. 2a). Csg2-GFP was functional in restoring starvation resistance and autophagy in Csg2-deleted yeast cells (Fig. S2a, b). We first confirmed that Csg2 was localized at ER membranes, as shown by the colocalization of Csg2-GFP with the ER marker HDEL protein (Fig. 2b). To clarify whether Csg2 is a calcium channel (the structure is shown in Fig. S2c), purified Csg2 protein (Fig. S2d) was reconstituted into a lipid bilayer and measured using a single-channel conductance assay[92–94] (Fig. 2c). A planar bilayer membrane was generated with a 150 μm-diameter aperture separating the cis-compartment and trans-compartment (Fig. 2c). After the addition of the Csg2 protein to the calcium chloride-containing buffer solution, a clear stepwise current increase was observed (Fig. 2d), indicating the incorporation of Csg2 proteins into the lipid bilayer membrane and its function in mediating ion flux. The control experiments (blank) without Csg2 protein showed no current increase (Fig. 2d). To evaluate the ion selectivity of Csg2, ion conductance assays were performed in buffers with other metal ions. The bar graph data clearly showed that Csg2 was strongly selective for calcium (Fig. 2e). To analyze whether the $Ca^{2+}$-channel Csg2 is regulated by calcium, in vitro channel conductance assays were performed in the presence of different concentrations of calcium. High-frequency ionic current noise (≥1 kHz) is correlated with dielectric noise and capacitive noise, mainly caused by low or loss of conductance of channel proteins incorporated into the lipid membrane. On the other hand, low-frequency fluctuation (≤1 kHz) is correlated with the conformational change and high conductance capability of the channel protein. The spectral density (amplitude (pA2/Hz)-frequency (Hz)) curve showed that the current trace of amplitude in the presence of low levels of calcium was different from that of the control solution in the low-frequency region, but not utterly significant; however, a high level of calcium caused the low-frequency region (≤1 kHz) to be greatly different from that of the control solution (Fig. S2e). These results indicated that the $Ca^{2+}$-channel Csg2 is regulated by high levels of calcium. All the above results demonstrated that Csg2 was a specific calcium channel. A genetic fluorescent probe of ER calcium, Stt3-jGCaMP7c (Fig. S2f), was generated by fusion of the ER membrane protein Stt3, which has been used to direct C-terminal aequorin to the ER lumen[95] with fluorescent jGCaMP7c, a genetically encoded calcium indicator (GECI) that has been designed to detect calcium[96]. Indeed, ER calcium levels were increased in Csg2-deleted

yeast cells, as shown by ER calcium measurements with the ER calcium probe Stt3-jGCaMP7c (Fig. S2f) and a fluorescence detection kit using a low-affinity calcium probe targeting ER calcium (Mag-Fluo-AM) (Fig. 2f), suggesting that Csg2 functions to mediate the efflux of calcium from the ER into the cytosol. Notably, the increased levels of ER calcium in Csg2-deleted yeast cells were less obvious when measured with the ER calcium probe Stt3-jGCaMP7c (Fig. S2f) than when measured with the ER calcium probe Mag-Fluo-AM (Fig. 2f). We speculate that the possible reason may be that jGCaMP7c has high contrast with a low baseline fluorescence compared with those of other jGCaMP7 indicators, such as jGCaMP7s (sensitive and slow), jGCaMP7f (fast kinetics) and jGCaMP7b (brighter baseline fluorescence)[96].

We next analyzed the involvement of the TM domains of Csg2 in its function in autophagy. The results suggested that all ten TM domains of Csg2 were important for the function of Csg2 in autophagy (Fig. S2g–i). There is an EF-hand domain (shown by the loop) between TM2 and TM3 of the Csg2 protein (Fig. S2c). The EF-hand domain contains approximately 40 amino acids and is involved in calcium binding[97–99]. The EF-hand domain (one to multiple) is usually located in calcium-modulating proteins such as calmodulin to induce their activation. By tagging the EF-hand domain of Csg2 with GFP and digesting it with protease K, we confirmed that the EF-hand domain of Csg2 was cytosolically distributed (Fig. S2j–m). To determine whether the EF-hand domain in Csg2 was involved in autophagy, Csg2 truncated with deletion of the EF-hand domain was constructed, and its function in autophagy was detected. The deletion of the EF-hand domain in Csg2 showed no adverse effect on autophagic function (Fig. S2n, o). The ER localization of Csg2 was not affected by TM or EF-hand domain deletion (Fig. S2p). Csg2 truncation mutants with the TM domain deleted could not restore autophagy in Csg2-deleted yeast cells (Fig. S2g–i); they could also not restore ER calcium homeostasis (Fig. S2q). In contrast, the EF-hand domain-deleted Csg2 truncation mutants restored autophagy (Fig. S2n, o) and could also restore ER calcium homeostasis in Csg2-deleted yeast cells (Fig. S2q). These data confirmed that the TM domains but not the EF-hand domain of Csg2 are required for its function in autophagy regulation.

## Autophagy blockade in Csg2-deleted cells is caused by increased calcium levels in the ER

We next analyzed how Csg2 deletion blocked autophagy. As shown above, Csg2 functions as a calcium channel (Fig. 2b-d); therefore, its deletion induced a calcium increase in the ER (Fig. 2f). We detected autophagy activities in Csg2-deficient yeast cells under high-calcium conditions (addition of calcium) and low-calcium conditions (addition of the ion-chelating agent EDTA or EGTA). The results showed that EDTA and EGTA, but not calcium, restored autophagy in Csg2-deficient yeast cells (Fig. 3a). The resistance to amino acid starvation was also restored in Csg2-deficient cells by the addition of EDTA or EGTA (Fig. 3b). As a control, Atg1 deletion caused autophagy blockade that

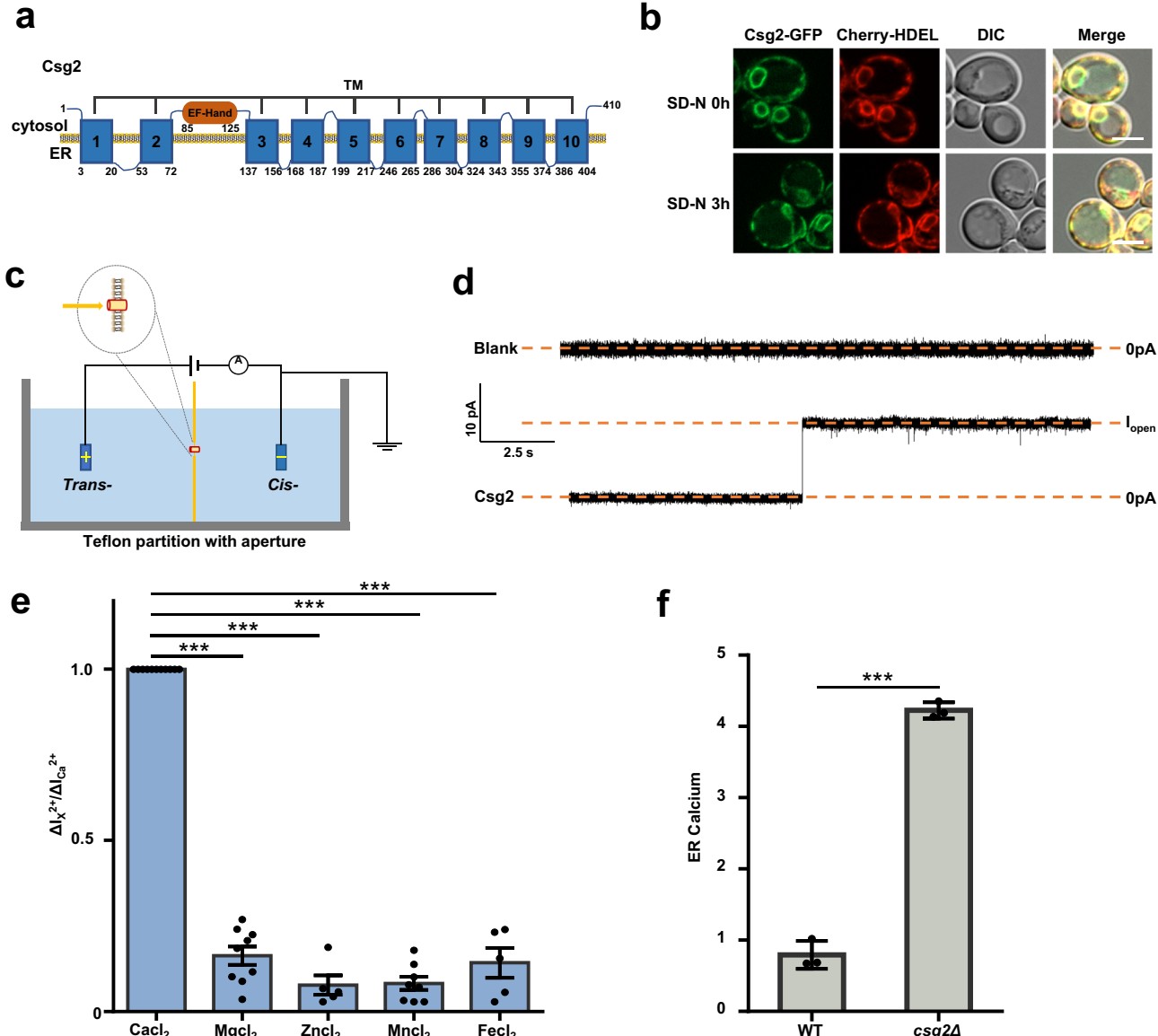

**Fig. 2 | Csg2 is an ER calcium channel for calcium release. a** Schematic representation of the domain organization of the Csg2 protein. Csg2 contains ten transmembrane domains and a potential calcium-binding EF-hand domain. **b** Colocalization of GFP-tagged Csg2 and ER marker Cherry-HDEL in WT yeast was detected using fluorescence assays before and after starvation in SD-N medium for 3 h. The experiment was repeated independently for three times with similar results. Representative images were shown. Scale bars: 5 μm. **c** Scheme illustration of measurement setup showing a protein molecule inserted into a 150-μm diameter patch of the lipid bilayer for channel conductance recording. **d** Current recording of the blank lipid bilayer (no protein added, top) and recording of the lipid bilayer with incorporated Csg2 protein (bottom) that shows channel conductance. The experiment was repeated independently at least three times with similar results

(Trans-: 30 mM $CaCl_2$, 10 mM HEPES, pH = 7.5 ± 0.05; Cis-: 300 mM $CaCl_2$, 10 mM HEPES, pH = 7.5 ± 0.05). **e** Histogram showing ion conductance change of the lipid bilayer with incorporated Csg2 protein after adding the indicated metal ions. The current change was standardized using the current generated by calcium chloride. Two-tailed t-test, n = independent experiments. $CaCl_2$: $n = 11$; $MgCl_2$: $n = 11$, $p = 4.80E{-}18$; $ZnCl_2$: $n = 6$, $p = 1.94E{-}08$; $MnCl_2$: $n = 8$, $p = 7.62E{-}17$; $FeCl_2$: $n = 5$, $p = 4.65E{-}08$. ***$p < 0.001$. Error bars, mean ± SEM. **f** ER calcium was more accumulated in csg2Δ cells compared to WT cells. WT yeast cells and csg2Δ cells were collected after starvation in SD-N medium for 3 h and ER calcium concentration was detected using Cell ER Calcium Concentration Fluorescence Detection Kit (see "Methods" for details). Two-tailed t-test, $n = 3$ independent experiments. $p = 1.2744E{-}05$. ***$p < 0.001$. Error bars, mean ± SD.

could not be restored by EDTA or EGTA (Fig. 3a, b), suggesting that the function of Csg2 in autophagy is specifically associated with its role in calcium regulation. The addition of calcium to WT cells, however, had no effect on autophagy and ER calcium levels (Fig. S3a, b), suggesting the capability of calcium homeostasis regulation in WT cells. Vacuolar transfer of the autophagic substrates GFP-Atg8 and RFP-Ape1 was restored by EDTA or EGTA in Csg2-deficient cells but not in Atg1-deficient cells (Fig. 3c, d). In addition to amino acid starvation-induced autophagy, glucose starvation-induced autophagy was also restored in

Csg2-deficient cells by EDTA or EGTA (Fig. S3c). The addition of calcium restrained the effect of EDTA or EGTA in restoring autophagy in Csg2-deficient cells (Fig. S3d). TPEN, a chelator with a relatively high affinity for zinc but a relatively low affinity for calcium, did not restore autophagy in Csg2-deficient cells (Fig. S3e). In contrast, BAPTA, a specific chelator of calcium, restored autophagy in Csg2-deficient cells (Fig. S3f). The ion chelators EDTA, EGTA, and BAPTA, which restored autophagy in Csg2-deleted yeast cells, also reduced the increased levels of ER calcium, while TPEN, which cannot restore autophagy, also

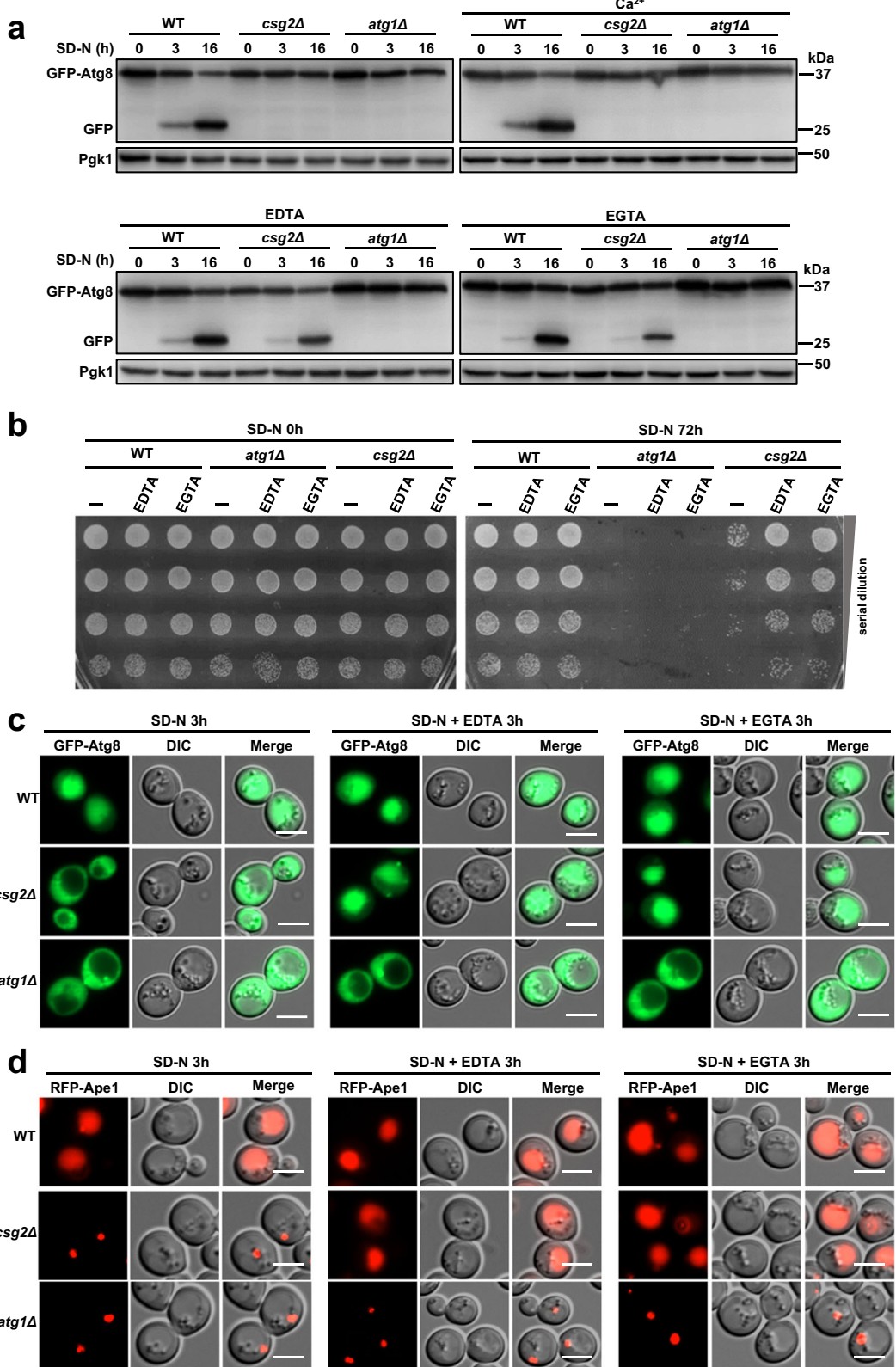

did not reduce the increased ER calcium in Csg2-deleted yeast cells (Fig. S3g, h). The dot-like compartments of the calcium probe in Csg2-deficient cells treated with calcium were colocalized or close to ER markers (Fig. S3i) when cell viability was not affected (Fig. S3j). Compared with the increased calcium in the ER caused by Csg2 deletion, the calcium in the vacuole was not affected by Csg2 deletion (Fig. S3k), suggesting that Csg2 functions specifically to regulate ER calcium.

These results suggested that an increase in calcium rather than a decrease in calcium was responsible for autophagy blockade in Csg2-deleted cells.

Based on the results that showed Csg2 as an ER-resident calcium channel, we hypothesized that Csg2 deletion caused calcium accumulation inside the ER, leading to autophagy blockade (schematic in Fig. 4a). We then tried to delete ER-resident Pmr1 (Fig. S4a), a

**Fig. 3 | The blockage of autophagy in Csg2 deleted cells was related with the accumulation of ER calcium. a** Degradation of autophagic substrate GFP-Atg8 in WT cells, csg2Δ cells, and atg1Δ cells after starvation in SD-N medium with or without CaCl₂ (100 mM), EDTA (5 mM), or EGTA (20 mM) was detected at the indicated times. The blots were probed with anti-GFP antibody and Pgk1 was used as a loading control. The samples derive from the same experiment and gels were processed in parallel. **b** WT cells, csg2Δ cells, and atg1Δ cells were checked for cell viability before and after starvation in SD-N medium with or without EDTA (5 mM) and EGTA (20 mM). The experiment was repeated independently for three times with similar results. **c** Vacuole transfer of autophagic marker GFP-Atg8 in csg2Δ cells was restored by addition of ion chelating agents EDTA and EGTA. Autophagic

marker GFP-Atg8 was investigated by fluorescence assays in WT cells, csg2Δ cells, and atg1Δ cells after starvation in SD-N medium with or without EDTA (5 mM) or EGTA (20 mM). The experiment was repeated independently for three times and representative images were shown. Scale bars: 5 μm. **d** Vacuole transfer of autophagic substrate RFP-Ape1 in csg2Δ cells was restored by addition of ion chelating agents EDTA and EGTA. Transportation of the autophagic substrate RFP-Ape1 was investigated by fluorescence assays in WT cells, csg2Δ cells, and atg1Δ cells after starvation in SD-N medium with or without EDTA (5 mM) or EGTA (20 mM). The experiment was repeated independently for three times and representative images were shown. Scale bars: 5 μm.

high-affinity P-type ATPase mediating calcium transport into the ER[100–103]. The WT, Csg2 deletion, and Csg2 Pmr1 double deletion cells expressing the ER calcium probe jGCaMP7c were investigated and the fluorescence intensity was quantified. The results showed that the increased fluorescence intensity caused by Csg2 deletion was reduced upon further deletion of Pmr1 (Fig. 4b, c), indicating that the increase in ER calcium caused by Csg2 deletion can be abolished by further Pmr1 deletion. We speculated that the overall content of calcium in the yeast cells is comparable but that the distribution within subcellular compartments (ER, for example) is different. This is supported by the experimental results. The total amounts of calcium in WT, Csg2 deletion, and Csg2 Pmr1 double deletion cells were measured, and the results showed no differences (Fig. S4b). During the analysis of autophagy activity, Pmr1 deletion clearly rescued autophagy in Csg2-deficient yeast cells, as shown by GFP-Atg8 processing (Fig. 4d), vacuolar transfer of Cherry-Atg8 (Fig. 4e) and starvation resistance (Fig. S4g). We then deleted the vacuolar calcium channel Pmc1, Vcx1, or Yvc1, and the results showed that autophagy was not restored (Fig. S4c, d). Furthermore, deletion of the cytosolic calmodulin-calcineurin-Crz1 signaling components Cna1, Cnb1, and Crz1[104–106] also did not rescue autophagy (Fig. S4e). These results suggested that the increase in ER calcium caused blockade of autophagy in Csg2-deleted cells.

The rescued autophagy and starvation resistance in Csg2-Pmr1 double-deletion yeast cells was blocked again upon Pmr1 re-expression (Fig. S4f, g). The D53A mutant of Pmr1, which is defective in transporting calcium[100], did not block autophagy in Csg2-Pmr1 double-deletion cells (Fig. 4f, g), which confirmed that calcium influx into the ER by Pmr1 was responsible for autophagy blockade in Csg2-deleted cells.

The above results suggested that the accumulated calcium inside the ER caused autophagy blockade in Csg2-deleted yeast cells. This was consistent with Csg2 functioning as a release channel of ER calcium (Fig. 2).

## Csg2 deletion disturbs sphingolipid metabolism, leading to increases in bioactive Sph and Cer levels

We next tried to determine the consequences of calcium accumulation in the ER caused by Csg2 deficiency that caused blockade of autophagy. First, we analyzed the transcription of autophagy genes in Csg2-deleted cells, as autophagy genes are generally induced under activation conditions such as amino acid starvation[107–110]. The results showed that the overall transcription or specific autophagy gene transcription was not significantly influenced by Csg2 deletion (Fig. S5a–c and Supplementary Data 1). We then performed a comprehensive lipidomics analysis[111–113] in WT and Csg2-deleted yeast cells under starvation conditions (schematic in Fig. 5a), as the ER is a main site for the synthesis of various lipids[60–62,114] and Csg2 regulates ER calcium. Principal-component analysis (PCA) showed that the cluster of WT yeast cells was clearly separated from the cluster of Csg2-deleted cells (Fig. S5d), and the two QC samples showed good consistency of signal during the mass spectrometric runs and good quality in the mass spectrometric data (Fig. S5e). Dysregulated lipid species were

quantified in samples derived from WT and Csg2-deleted yeast cells (Fig. 5b and Supplementary Data 2 and 3). Among the lipids analyzed, bioactive sphingolipids, phytoCers, and PHS exhibited the largest and most significant differences between WT and Csg2-deleted cells (Fig. 5b, c and Fig. S5f). In yeast cells, phytoCers and PHS are immediate precursors of complex sphingolipids (Fig. 5d). The levels of the complex sphingolipid mannosylinositol phosphorylceramide (MIPC) were also reduced in Csg2-deleted yeast cells under starvation conditions (Fig. S5g). The increase in sphingolipid PHS in Csg2-deleted yeast cells was not found when cells were cultured in nutrient-rich conditions (Fig. S5h).

These results suggest that the disturbed metabolism of sphingolipids in Csg2-deleted cells under starvation conditions, especially the increases in the levels of bioactive sphingolipids, may be responsible for the autophagy blockade induced by calcium accumulation in the ER.

## Increased PHS mediates autophagy blockade induced by accumulation of ER calcium in Csg2-deleted yeast cells

Sphingolipids play essential roles not only as structural components of membranes but also as bioactive signaling molecules[60–63]. Many key enzymes and regulators of sphingolipid metabolism were first discovered in yeast, and based on the high degree of conservation, various mammalian homologs have been identified[65,66]. Cer is the basic structural unit of sphingolipids and is formed through the conjugation of a very long-chain fatty acid to backbone Sph (schematic in Fig. S6a). The ER is the main site for sphingolipid synthesis in cells (schematic in Fig. S6b). Given the observation that calcium accumulated in the ER and that phytoCer and PHS levels were increased in Csg2-deleted yeast cells, we hypothesized that deletion of the calcium channel Csg2 would cause a calcium increase in the ER, thus disturbing sphingolipid metabolism and leading to increased bioactive sphingolipid levels, eventually blocking autophagy (schematic in Fig. 6a). Three temperature mutants of key enzymes in sphingolipid synthesis were then used to test this hypothesis (Fig. 6b, c). The results showed that under nonpermissive temperature (enzyme dysfunctional) conditions, mutation of Lip1 and Aur1[115–118] caused blockade of autophagy, while mutation of upstream Tsc10[115,116,119,120] did not affect autophagy (Fig. 6b). This indicated that the increase in dihydrosphingosine (DHS) or PHS caused blockade of autophagy (Fig. 6c), which is consistent with the observation of increased PHS in Csg2-deleted cells (Fig. 5a–c). To further test this hypothesis, we analyzed whether the blocked autophagy in Csg2-deleted yeast cells could be rescued by reducing DHS and PHS levels with the Tsc10 mutation in Csg2-deleted cells. The results showed that the Tsc10 mutation, but not the Lip1 or Aur1 mutation, restored autophagy in Csg2-deleted yeast cells (Fig. 6d, e). This confirmed that the increased DHS or PHS levels in Csg2-deleted yeast cells blocked autophagy. Fumonisin B1, an inhibitor of ceramide synthase, caused autophagy blockage (Fig. S6c). Deletion of the MIPC synthase Sur1/Csh1, caused autophagy blockage and an increase in PHS (Fig. S6d, e). The temperature-sensitive mutants combined with Csg2 deletion showed no effect on cell viability after starvation at the time points when autophagy was detected (Fig. S6f). The PHS levels in

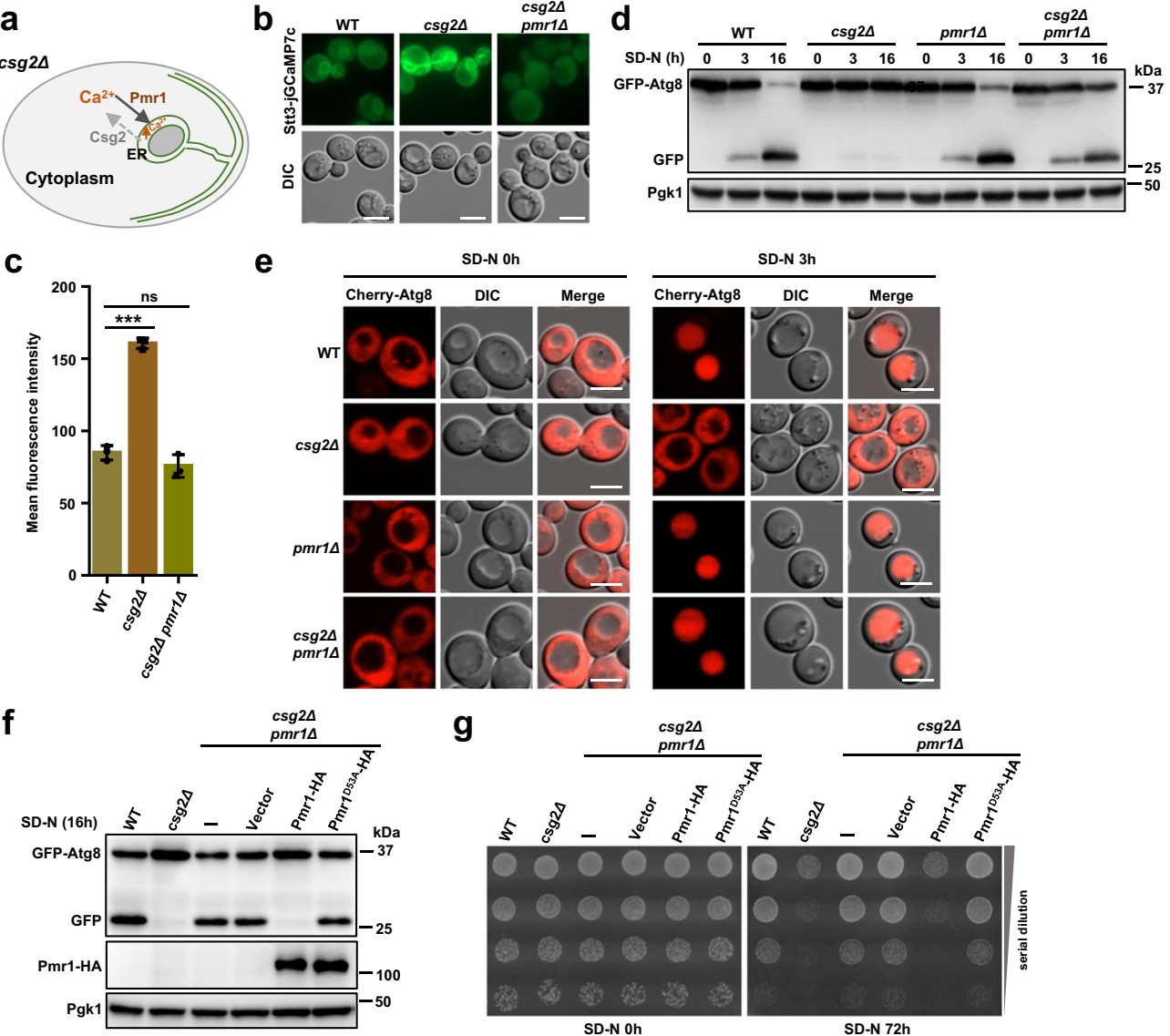

**Fig. 4 | Calcium accumulation in ER caused blockage of autophagy in Csg2 deleted yeast cells. a** Schematic diagram of ER calcium accumulation in csg2Δ cells. When the transport directions of calcium (into the ER by Pmr1 and out of the ER by Csg2) was disrupted by Csg2 deletion, calcium levels were increased in the ER lumen. **b, c** ER calcium probe Stt3-jGCaMP7c were investigated by fluorescence assays in indicated cells after starvation in SD-N medium for 3 h. Representative parameters were set the same to compare the signal intensities. Representative images (**b**) and fluorescence intensity quantification (**c**) were shown. Scale bars: 5 μm. Two-tailed t-test, n = 3 independent experiments. p values: 2.902E-05, 0.1626. ***p < 0.001. ns: p > 0.05, not significant. Error bars, mean ± SD. **d** WT cells and csg2Δ cells with or without deletion of ER calcium channel Pmr1 were analyzed for the autophagic degradation of GFP-Atg8 after starvation in SD-N medium. The blots were probed with anti-GFP antibody and Pgk1 was used as a loading control. The

samples derive from the same experiment and gels were processed in parallel. **e** Autophagic marker Cherry-Atg8 was expressed and investigated by fluorescence examination in WT cells, csg2Δ cells, pmr1Δ cells, and csg2Δpmr1Δ cells before and after starvation in SD-N medium. The experiment was repeated independently for three times and representative images were shown. Scale bars: 5 μm. **f** WT or D53A mutant Pmr1 was expressed in csg2Δpmr1Δ cells and the degradation of GFP-Atg8 after starvation in SD-N medium was analyzed. The blots were probed with anti-GFP antibody and HA antibody, Pgk1 was used as a loading control. The samples derive from the same experiment and gels were processed in parallel. **g** Starvation resistance conferred by autophagy in csg2Δpmr1Δ cells was prohibited by exogenous expression of Pmr1 but not by D53A mutant of Pmr1 that lose calcium channel ability. The experiment was repeated independently for three times with similar results.

these temperature-sensitive mutant cells (Fig. S6g) were in line with the autophagy activities (Fig. 6e). We then investigated which specific subclass of sphingolipids, DHS or PHS, was responsible for autophagy blockage. 3-KDS is reduced by Tsc10 to form DHS, which can be further converted into PHS by the hydroxylase Sur2[121]. The two long-chain bases (LCBs), DHS and PHS, can be phosphorylated by two kinases, Lcb4 and Lcb5[122], to form LCB-phosphates (LCBPs), DHS-1p and PHS-1P (schemed in Fig. 6f, upper). The results showed that further deletion of Sur2, but not Lcb4 or Lcb5, in Csg2-deleted yeast cells restored autophagy, as shown by analyses of GFP-Atg8 processing (Fig. 6f, g),

starvation resistance (Fig. 6h) and vacuole transfer of Cherry-Atg8 (Fig. S6h). Re-expression of Sur2 blocked autophagy again in Csg2-Sur2 double-deletion cells (Fig. 6g, h). In addition to Sur2, there are two other hydroxylases, Scs7 and Ccc2[121,123–126], that modify Cers by hydroxylation (schematic in Fig. S6b). However, further deletion of Scs7 or Ccc2 in Csg2-deficient cells did not restore autophagy (Fig. S6i, j). Deletion of Sur2 in Csg2-deleted yeast cells caused a decrease in PHS (Fig. S6k). Furthermore, deletion of Sur2 in Sur1/Csh1-deleted yeast cells restored autophagy (Fig. S6l). Deletion of proteins at the other positions of the sphingolipid metabolism process, Rsb1 and Ypc1, in

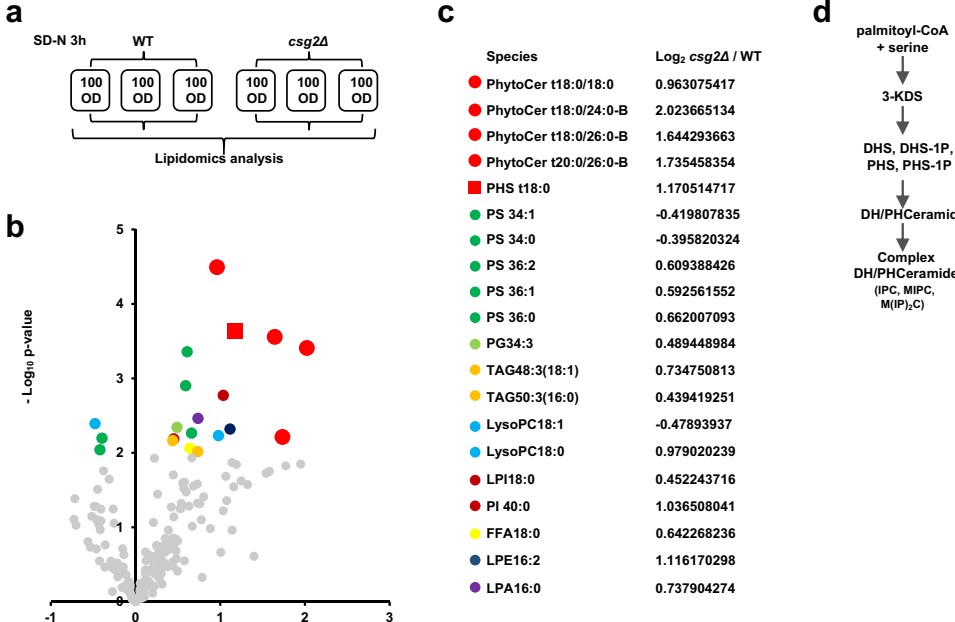

**Fig. 5 | Bioactive sphingolipids phytosphingosine and phytoceramide accumulated in csg2Δ cells. a, b** Phytosphingosine and ceramide accumulated in csg2Δ cells after starvation. 100OD WT cells and csg2Δ cells were collected after 3 h of starvation in SD-N medium and subject to lipidomics analysis. Experiments were set up in three biologically independent samples (**a**). The mostly increased lipids with

high confidences were found to be bioactive sphingolipids, phytoceramides (PhytoCer) (colored in red cycles), and phytosphingosine (PHS) (colored in red square). Analyzed with t-test, $n = 3$ biologically independent samples. **c** The ratios of lipids that were obviously increased in Csg2 deleted cells. **d** Brief schematic diagram of the process of sphingolipid synthesis in yeast.

Lcb4/5-deleted cells, showed no adverse effect on autophagy (Fig. S6m). The above results support the hypothesis that calcium accumulation in the ER induced by Csg2 deletion disrupts sphingolipid metabolism and causes increases in PHS levels, which eventually block autophagy (schematic Fig. 6i). Can adding PHS restore the blockade of autophagy? The results showed that the addition of PHS to Csg2-Sur2 double-deletion cells, Csg2-Pmr1 double-deletion cells and Csg2 deletion plus EDTA/EGTA-treated cells caused blockade of autophagy (Fig. 6j-l). The addition of PHS caused an increase in intracellular PHS in Csg2-Sur2 double-deletion cells but not in WT cells (Fig. S6n, o), in line with the results that added PHS inhibited autophagy in Csg2-Sur2 double-deletion cells but not in WT cells (Fig. 6j). The blockage of autophagy seemed to be specific, as cell viability was not affected at the time of autophagy detection (Fig. S6p, q). These results suggested that the increase in PHS induced by calcium accumulation in the ER blocked autophagy in Csg2-deleted cells. Using the autophagy marker GFP-Atg8, we found that added PHS inhibited autophagosome formation, as shown by the diffuse cytosolic distribution of GFP-Atg8 (Fig. S6r). Added PHS did not affect the vacuole transfer and degradation of endocytosis protein Sna3 (Fig. S6s, t), indicating that PHS inhibition of autophagy was specific. Furthermore, deletion of Lcb3, a long-chain base-1-phosphate phosphatase, showed no adverse effect on autophagy in the presence of added PHS (Fig. S6u, v), indicating that phosphorylated DHS or PHS did not regulate autophagy.

Together, these findings suggest that deletion of Csg2, the ER-resident calcium channel, causes accumulation of calcium in the ER, which disturbs sphingolipid metabolism, leading to increases in PHS that block autophagy (schematic in Fig. S6w).

### Increases in calcium caused by Csg2 deletion reduce the protein stability of Aur1, a Cer phosphoinositol transferase required for complex sphingolipid synthesis

We next tried to determine how Csg2 deletion induced calcium accumulation in the ER and caused an increase in PHS. One speculation was that certain enzymes in the sphingolipid synthesis

pathway were dysfunctional in Csg2-deleted cells (schematic in Fig. 7a, upper). Given that both Lip1 mutation and Aur1 mutation can cause increases in PHS and that both Lip1 mutation and Aur1 mutation can block autophagy (schematic in Fig. 7a, lower), we speculated that the functions of Lip1 or Aur1 may be disrupted in Csg2-deleted yeast cells. The mRNA levels of Lip1, Aur1 and other enzymes in the sphingolipid synthesis pathway were not affected by Csg2 deletion (Fig. S7a), similar to the observation that Csg2 did not regulate the transcription of autophagy factors (Fig. S5a–c). We then detected the protein levels of Aur1 in WT and Csg2-deleted cells, and interestingly, the results showed that the Aur1 protein level was dramatically reduced in Csg2-deleted cells, while Tsc10 and Lip1 levels were not affected (Fig. 7b). The reduced protein levels of Aur1 by Csg2 deletion were still observed when the position of the GFP tag was changed from the C-terminus to the N-terminus of Aur1 (Fig. S7b). Aur1 forms a complex with Kei1[127–130], a regulatory subunit, to catalyze the synthesis of complex sphingolipids (IPCs). The interaction between Aur1 and Kei1 was dramatically reduced in Csg2-deleted yeast cells (Fig. 7c). This was probably caused by disrupted Aur1, as Kei1 showed normal protein levels (Fig. S7c).

We then investigated whether the Aur1 abnormality in Csg2-deleted yeast cells was caused by increased calcium levels. In the presence of the chelating agents EDTA and EGTA, the Aur1 protein levels in Csg2-deleted cells were maintained (Fig. 7d). Deletion of Pmr1 restored the protein levels of Aur1 (Fig. S7d), indicating that the accumulation of calcium in the ER that caused disruption of Aur1. Furthermore, double deletion of Csg1 and Csh1 showed no effect on the protein levels of Aur1 (Fig. S7e). We speculated that although both Csg2 deletion and Csg1/Csh1 deletion caused blockage of autophagy, the underlying mechanisms are different. Deletion of Csg2, the ER calcium channel, induced the accumulation of calcium in the ER, leading to disruption of Aur1 and a subsequent increase in PHS that caused blockage of autophagy. Deletion of Csg1/Csh1, the enzymes in the sphingolipid synthesis pathway, induced an increase in PHS and further blocked autophagy. The disruption of Aur1 was specific, as

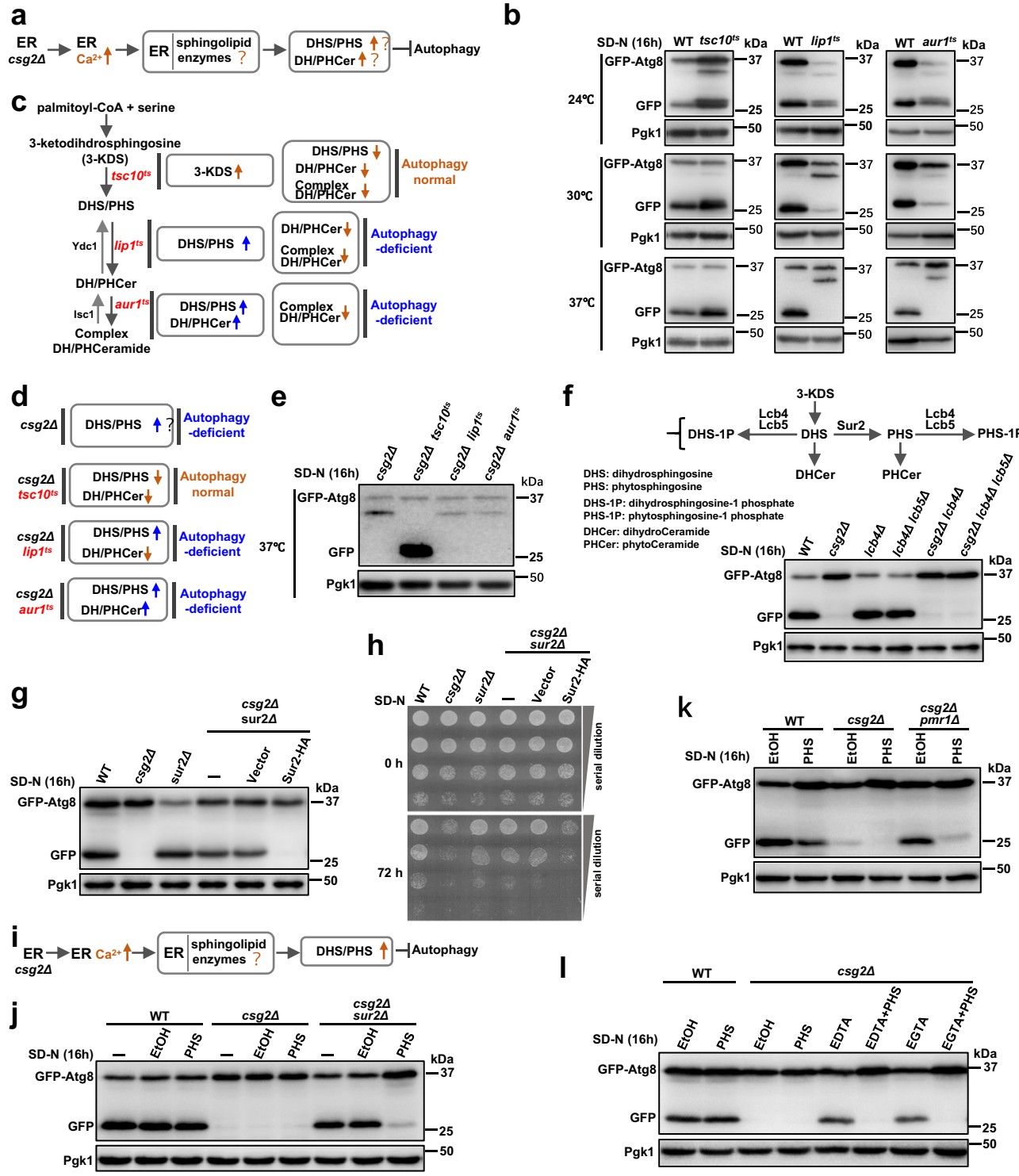

other proteins in the sphingolipid synthesis process, such as Kei1, Tsc10, Lip1, Sur2, Sur1, and Csh1 were not affected by Csg2 deletion (Figs. 7b, S7c, f–h). The protein levels of Aur1 in Csg2-deleted cells were restored by the proteasome inhibitor MG132 but not by the vacuole inhibitor NH4Cl (Fig. S7i, j), suggesting that Aur1 was subject to proteasome degradation in Csg2-deleted cells. We speculated that such degradation was efficient, as even when overexpressed in the high-copy vector, the protein levels of Aur1 were low in Csg2-deleted cells under starvation conditions, which was in line with the results that autophagy was not recovered (Fig. S7k). We then analyzed Aur1 protein levels in WT, *csg2Δsur2Δ*, or *csg2Δpmr1Δ* cells. The results showed that

the protein levels of Aur1 were normal in WT cells and in *csg2Δpmr1Δ* cells, and the addition of PHS showed no effect (Fig. S7l). The protein levels of Aur1 were disrupted in *csg2Δsur2Δ* cells, and the addition of PHS showed no effect (Fig. S7l). We speculated that the protein levels and localization of Aur1 were regulated by ER calcium but not PHS. We analyzed Tor complex 1 (TORC1) activity in *csg2Δ* cells (PHS increased) by detecting the activation (phosphorylation) of its downstream protein Sch9 (ortholog of mammalian S6 kinase). The results showed that phosphorylation of Sch9 was increased in *csg2Δ* cells under starvation conditions and this increase was abolished by further deletion of Sur2 (Fig. S7m). This was in line with results showing that under starvation

**Fig. 6 | Accumulated phytosphingosine caused blockage of autophagy in csg2Δ cells. a** Hypothesis of mechanism of increased calcium in ER caused autophagy blockage through disturbing sphingolipid synthesis pathway. **b** WT and temperature sensitive tsc10ts, lip1ts, and aur1ts yeast cells were checked by GFP-processing assays using GFP-Atg8 after starvation in SD-N medium for 16 h at permissive 24 °C and nonpermissive 30 °C or 37 °C conditions. The blots were probed with anti-GFP antibody and Pgk1 was used as a loading control. The samples derive from the same experiment and gels were processed in parallel. **c** Schematic diagram showed effects of mutations of sphingolipid synthesis enzymes Tsc10, Lip1, and Aur1 on autophagy. **d, e** GFP-processing assays using GFP-Atg8 were checked in tsc10ts, lip1ts, and aur1ts yeast cells combining with Csg2 deletion after starvation in SD-N medium for 16 h at nonpermissive 37 °C. The blots were carried out as in (**b**). **f** Autophagic degradation of GFP-Atg8 in indicated yeast cells was analyzed after

starvation in SD-N medium for 16 h. The blots were carried out as in (**b**). **g** Autophagic degradation of GFP-Atg8 in indicated yeast cells was analyzed after starvation in SD-N medium for 16 h. The blots were carried out as in (**b**). **h** Cell viability of indicated yeast cells was analyzed before and after starvation in SD-N medium for 72 h. The experiment was repeated independently for three times with similar results. **i** Hypothesis of mechanism of increased calcium in ER caused blockage of autophagy. **j, k** Autophagic degradation of GFP-Atg8 was analyzed in indicated yeast cells after starvation in SD-N medium with addition of PHS (5 µM, solved in EtOH) or control EtOH for 16 h. The blots were carried out as in (**b**). **l** Autophagic degradation of GFP-Atg8 was analyzed in indicated yeast cells after starvation in SD-N medium with addition of PHS (5 µM, solved in EtOH) or control EtOH at the presence of EDTA (5 mM) or EGTA (20 mM) for 16 h. The blots were carried out as in (**b**).

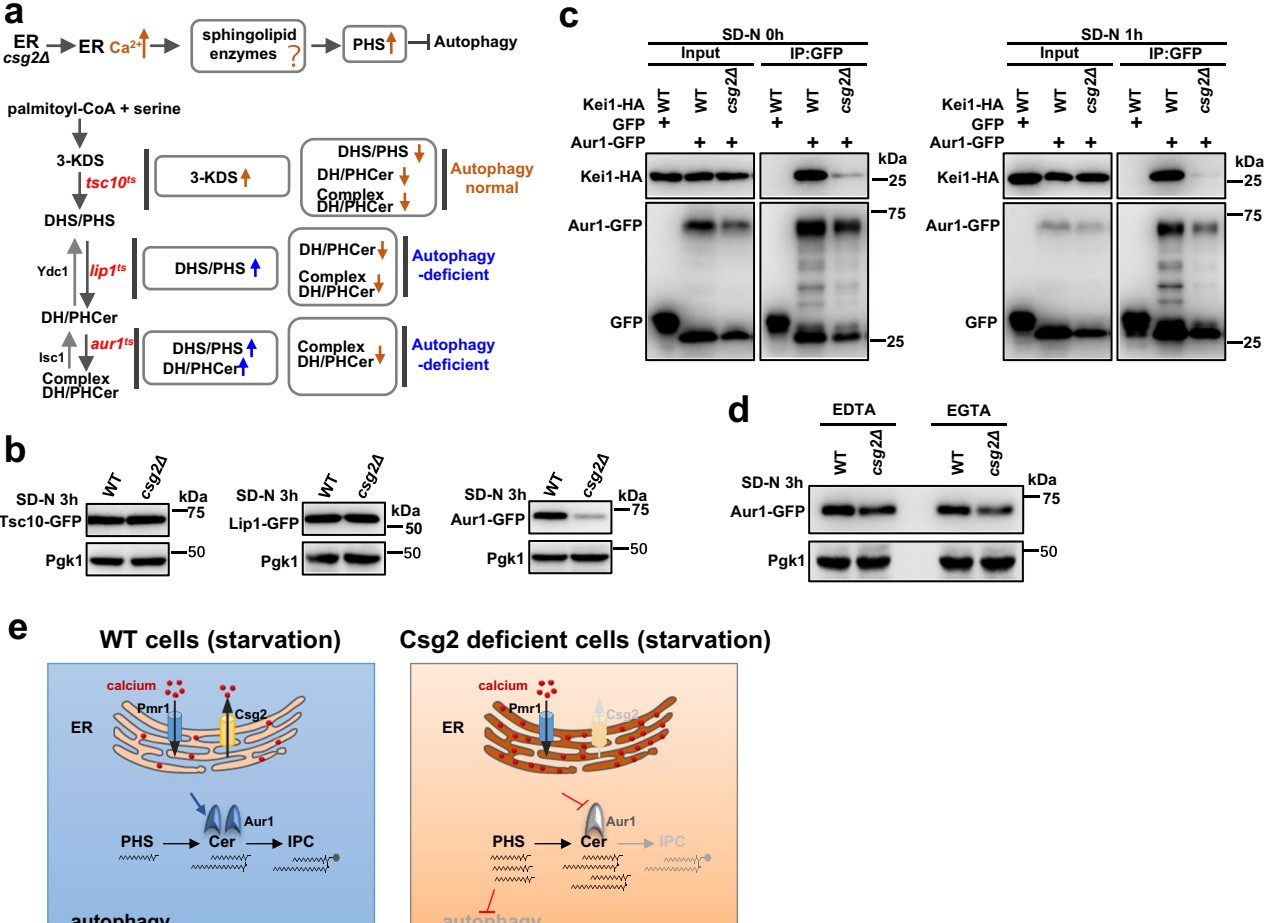

**Fig. 7 | Complex ceramide synthase Aur1 was disrupted by accumulated ER calcium in csg2Δ cells. a** Upper, schematic diagram of the hypothesis of calcium accumulation in ER caused disruption of enzymes involved in sphingolipid biosynthesis, leading to increased levels of PHS and blockage of autophagy. Lower, schematic diagram of the effects of mutation in Tsc10, Lip1, and Aur1 on autophagy. **b** Tsc10-GFP, Lip1-GFP, and Aur1-GFP were expressed in WT cells and csg2Δ cells after starvation in SD-N medium and their proteins levels were analyzed. The blots were probed with anti-GFP antibody and Pgk1 was used as a loading control. The samples derive from the same experiment and gels were processed in parallel. **c** The interaction between Aur1 and Kei1 was weakened by Csg2 deletion. WT cells and csg2Δ cells with expression of Aur1-GFP and Kei1-GFP were collected before and after starvation in SD-N medium and subject to Co-IP assays. The samples

derive from the same experiment and gels were processed in parallel. **d** Aur1-GFP was expressed in WT cells and csg2Δ cells and its protein levels were analyzed after starvation in SD-N medium with EDTA (5 mM) or EGTA (20 mM). The blots were probed with anti-GFP antibody and Pgk1 was used as a loading control. The samples derive from the same experiment and gels were processed in parallel. **e** Schematic diagram of the integration of ER calcium homeostasis with sphingolipid metabolism and autophagy by calcium channel Csg2. Csg2 is localized at the ER membranes and functions as a calcium release channel maintaining the calcium homeostasis in ER. When Csg2 is absent, abnormally high levels of calcium accumulates in ER to cause the disruption of complex ceramide (IPC) synthase Aur1, which results in increase of bioactive sphingolipid PHS that inhibits autophagy.

conditions PHS was increased in *csg2Δ* cells and this increase was abolished by further deletion of Sur2 (Fig. S6k). These results suggested that TORC1 activity was activated by increased PHS in *csg2Δ* cells, and TORC1 activation inhibits autophagy.

These results suggest that the increases in calcium levels caused by Csg2 deletion are responsible for the disruption of Aur1, leading to disturbed sphingolipid metabolism and increased bioactive PHS levels that eventually inhibit autophagy.

## Discussion

Here, we integrated calcium homeostasis with sphingolipid metabolism and autophagy (Fig. 7e). As one of the most common ions functioning in cells, calcium has been widely studied. The storage and functions of calcium in eukaryotic cells are highly conserved. Less is known about the consequences and underlying mechanism of excessive accumulation of calcium in reservoirs such as the ER. We found in yeast that the accumulation of calcium in the ER caused by deletion of Csg2, an ER-resident calcium channel, disturbed sphingolipid metabolism, leading to an increased amount of bioactive PHS and eventually blocking autophagy (Fig. 7e). This result suggests that the stored calcium in intracellular reservoirs not only acts as a backup for calcium but also has biological consequences. The ability of Csg2 to act as a calcium channel increased with higher concentrations of calcium, suggesting that Csg2 mediates calcium flux out of the ER with increased efficiency when there are abnormally high levels of calcium in the ER. It makes sense to maintain the normal steep gradient of calcium in the ER; Csg2 is restrained to prevent calcium from exiting the ER and functions efficiently when the concentration of calcium in the ER exceeds the physiological level.

However, autophagy is not inhibited in WT cells when calcium or PHS is added. We speculated that WT cells can resist added calcium by maintaining the intracellular calcium homeostasis, which means that ER calcium in WT cells is not affected when calcium is added. Various calcium channels may contribute to the calcium homeostasis. We speculated that the addition of PHS to WT cells may not cause a significant change in intracellular PHS levels; therefore, the autophagy in WT cells was not inhibited when PHS was added. We then experimentally investigated the PHS levels in WT cells when PHS was added, and the results showed that the intracellular PHS levels were not affected by the addition of PHS (Fig. S6n). We speculated that the possible reason is that the sphingolipid metabolic enzymes in WT cells function to promote the efficient metabolism of added PHS, leading to steady levels of intracellular PHS. This was supported by the results showing that added PHS caused a significant increase in intracellular PHS in *csg2Δ sur2Δ* cells (Fig. S6o), in which situation the PHS upstream enzyme Sur2 was deleted and the downstream enzyme Aur1 was disrupted by accumulated ER calcium caused by Csg2 deletion.

The mechanism by which Csg2-mediated calcium homeostasis regulates autophagy needs to be verified to determine whether it is conserved in mammalian cells. Of note, the roles of calcium in autophagy are rather controversial in mammalian cells, with many studies showing an inhibiting effect of calcium on autophagy, contrasting with the stimulating action of calcium shown in other studies[5]. Overall analysis of these studies showed that these controversies come from complex intracellular calcium signaling[5]. The eventual effects of calcium on autophagy greatly rely on spatiotemporal characteristics and the amplitude of calcium signals, the cell growth conditions, the tissue characteristics of cells, and the types of autophagy (basal or selective autophagy)[5]. Although it seems that many parameters can determine the stimulatory or inhibitory effects of calcium on autophagy, calcium essentially regulates autophagy. As cytosolic and organellar calcium homeostasis is regulated by calcium channels, the illustration of correlations between calcium channels and autophagy activities under various situations will help discover the calcium-dependent mechanisms of autophagy regulation.

The consequence of calcium accumulation in the ER is disruption of sphingolipid metabolism through dysfunction of Aur1, a Cer phosphoinositol transferase that catalyzes Cer formation into complex Cer. At present, it is unclear how increased calcium levels in the ER affect the protein stability and cellular localization of Aur1. This effect is specific, as the regulatory molecule Kei1 that functions together with Aur1 and other enzymes involved in sphingolipid synthesis, such as Tsc10 and Lip1, were found to be unaffected. It is possible that increased calcium levels in the ER cause overwhelming binding with Aur1, disrupting the protein folding of Aur1 and leading to subsequent degradation and mislocalization. Another possibility is that the increased calcium in the ER competes with and prevents certain ions from binding to Aur1, which causes disruption of the structure of Aur1. It will be interesting to explore the interplay between calcium and Aur1 in the future.

Our study clarifies that a specific kind of sphingolipid, PHS, inhibits autophagy. Sphingosines, DHS, and PHS are single-chain sphingolipids that not only serve as the backbone of complex sphingolipids but also possess important biological signaling activities[65,131–133]. They play essential roles in cellular functions such as cell proliferation, apoptosis, migration, inflammation, and intracellular trafficking and play roles in diseases such as cancer and Niemann-Pick disease type C (NPC)[65,133–137]. Although we established the link between PHS and autophagy inhibition at the autophagosome formation stage, how this inhibition is achieved is only slightly uncovered. We speculate that PHS, as a bioactive sphingolipid, may regulate TORC1, which suppresses autophagy and promotes cell growth under nutrient-rich conditions[138–140]. Considering that PHS and free fatty acids (FFAs) form Cers and that TORC1 promotes the synthesis of FFAs through the Sit4-GSK3-El2/3 signaling axis[65], it is possible that the increase in PHS levels activates TORC1 to increase the production of FFAs, resulting in enhanced Cer formation. This mechanism is supported by the observation that FFAs and Cers were both increased in Csg2-deleted cells (Fig. 5a–c). Therefore, PHS may represent a nutrient signal that delivers growth information to TORC1, which leads to TORC1 activation and subsequent suppression of autophagy. This possibility is supported by the observation that the localization of the autophagosome-initiating factors Atg1 and Atg13, downstream factors inhibited by TORC1[77,78], was disturbed at the phagophore assembly site (PAS) (data not shown). TORC1 suppresses autophagosome initiation through phosphorylation of Atg13, preventing activation of the initiating Atg1-Atg13 kinase complex[77,78]. Thus, it is possible that PHS activates TORC1 to suppress Atg1-Atg13, which blocks autophagosome initiation and causes autophagy inhibition. Another possibility is that PHS may directly disturb membrane resources and the trafficking of membrane lipids and membrane-binding Atg proteins such as Atg18 for autophagosome formation. The exact mechanistic details of PHS-mediated inhibition of autophagy remain to be elucidated. Therefore, additional investigation is required to clarify the role of PHS in the regulation of autophagy, which will promote the discovery of additional targets of bioactive sphingolipids in the modulation of cellular functions. Intact and impaired sphingolipid-related autophagy is significantly involved in various pathological processes and diseases[141–143], including neurodegenerative diseases such as Alzheimer's disease[144–146], Batten disease[147] and Niemann-Pick disease[111]. Thus, a better knowledge of the roles of sphingolipids in autophagy regulation will provide useful insights for both basic and translational studies, and the unique crosstalk between sphingolipids and autophagy is an exciting area for further research.

Recently, it was found that the metazoan-specific protein EPG-4/EI24 regulates calcium transients on the outer surface of the ER membrane, which induces phase separation of the autophagy protein FIP200 for the initiation of autophagosome formation[144]. Combined with our study here, we speculated that increased calcium functions in a multifaceted manner, depending on the venues where it is located, concentration on the outer surface ER or accumulation inside the ER. When calcium is concentrated on the outer surface of the ER, it promotes autophagy as a "gas pedal" for autophagy activation; however, when calcium is accumulates in the lumen of the ER, it inhibits autophagy as a "brake pedal" for autophagy inhibition. It will be interesting to investigate the correlation and evolution of these two mechanisms of autophagy regulation by calcium in different eukaryotic organisms.

In conclusion, our study provides insights into how calcium homeostasis in the ER maintained by the Csg2 channel modulates sphingolipid metabolism and autophagy. In light of the widespread and conserved roles of calcium, sphingolipids and autophagy in eukaryotic organisms, their integration established here may be conserved among eukaryotes in particular physiological contexts.

## Methods

### Yeast strains and manipulations

All yeast strains and their genotypes and plasmids used in this study are listed in Supplementary Tables S1 and S2. Yeast strains of gene deletion were generated by PCR-based gene knockout strategy using HIS3MX6, hphNT1, or natNT2 selection antibiotic markers. Gene specific products from PCR amplification of HIS3MX6, hphNT1, or natNT2 were transformed into yeast cells using herring sperm DNA in Li-Ac containing PEG3350 buffer. Antibiotic selection was achieved using histidine deficient medium, 250 µg/L hygromycin B (Invitrogen) or 250 mg/mL Zhongshengmycin (Klbios). Gene deletions were confirmed by PCR using crude DNA extracts. Except for the temperature-sensitive strains, Yeast cells were cultured using standard conditions and media: cells were cultured overnight at 30 °C in YPD-media containing Glucose (2%), Yeast Extract (1%), and Bacteriological Peptone (2%), then were inoculated in same medium cultured to log phase. Autophagy was induced by nitrogen starvation: cells were cultured in YPD medium to the logarithmic stage, and then were replaced to SD-N medium containing 0.17% yeast nitrogen base without amino acids and ammonium sulfate and 2% glucose or SD-Glu medium containing amino acids and 0.17% yeast nitrogen base for indicated hours.

### Yeast strains starvation, GFP-cleavage, and cell viability assays

Yeast cells were cultured in YPD medium until log phase, and then switched to lacking nitrogen medium for indicated times. For GFP-cleavage assay, which was measured by monitoring the accumulation of the released GFP moiety. GFP-Atg8 or GFP-50Q of the autophagy substrate was degraded in vacuoles to produce the free GFP moiety that is highly stable and resists degradation in vacuoles during starvation. For cell viability determination assay, yeast cells were cultured in SD-N medium for indicated days and then spotted on YPD-plate with gradient dilution for re-growth with serial dilutions.

### Yeast temperature-sensitive strains starvation and GFP-cleavage assays

Temperature-sensitive yeast cells were cultured overnight at 24 °C in YPD medium, then inoculated to the same medium until log phase at 24 °C, and then switched to SD-N medium at 24 °C, 30 °C, and 37 °C for indicated hours. In the GFP-cleavage assays, GFP-Atg8 as the autophagy substrate was measured by monitoring the accumulation of the released GFP moiety.

### Immunoblotting analyses

The indicated cells were collected and cleaved with HU lysis buffer at 65 °C for 20 min to extract total proteins. Protein samples were resolved on SDS-PAGE gels and performed with specific antibodies and secondary anti-mouse or anti-rabbit antibodies conjugated to horseradish peroxidase (HRP) (1:5000). Images were achieved with chemiluminescence and taken by Image Lab (BioRad). Mouse monoclonal antibodies to against HA-epitope (sc-7392; 1:2000 for IB) and GFP (sc-9996; 1:5000 for IB) were purchased from Santa Cruz Biotechnology, mouse monoclonal antibody to Pgk1 (ab113687; clone 22C5D8; 1:10,000 for IB) was purchased from Abcam. Mouse monoclonal antibody to Cherry (A2080; 1:5000 for WB) was purchased from Abbkine. Rabbit polyclonal antibody to Ape1 (1:1000 for IB) was a gift from Dr. Cong Yi (Zhejiang University). Rabbit monoclonal antibodies to p-Sch9 (Cat# 4858; 1:2000 for IB) and Sch9 (Cat# 2217; 1:1000 for IB) were purchased from Cell Signaling Technology. All images were

presented as representative results for at least three replicates in the paper.

### Fluorescence microscopy observation assays

For observation of fluorescent tag labeled proteins, Yeast cells transformed into associated plasmids were cultured to log phase in synthetic complete medium (0.17% YNB, 2% glucose, and 0.2% synthetic amino acids except for the deletion of amino acid) at 30 °C. Cells were washed twice using distilled water and switched in SD-N medium for indicated different times. Aliquots of liquid culture were then collected at indicated time points and allowed to precipitate on concanavalin A (Solarbio, C8110) coated cover glass for 5 min. Images were captured by a fully automated Zeiss inverted microscope (Observer 7) with Apotome. images were analyzed using Zeiss ZEN and Image J. All images were presented as representative results for at least three replicates in the paper.

### Purification of Csg2

$2.5 \times 10^6$ Sf9 cells transformed plasmid pFastBac-Dual-Csg2-TEV-2*Strep-Flag were collected after cultured 48 h. Then cells were crushed in lysis buffer (25 mM HEPES7.4, 200 mM NaCl, 10% glycerol, 0.5 mM Tcep, 800 nM Apro, 5 µg/ml Leu,0.2 mM AEBSF) for 40 times on the ice. The sediment was resuspended by lysis buffer with 1% DDM and 0.2% CHS after centrifuged $3200 \times g$/4 °C for 40 min and incubated 4 °C overnight. The supernatant collected after supercentrifuged $140,000 \times g$/4 °C for 30 min was incubated with Strep-Tactin beads (iBA) for 40 min at 4 °C. After centrifugation, beads were washed with wash buffer (25 mM HEPES7.4, 200 mM NaCl, 5% glycerol, 0.5 mM Tcep, 800 nM Apro, 5 µg/ml Leu, 0.2 mM AEBSF, 0.05% DDM + 0.01% CHS), then eluted with elute buffer (100 mM Tris-HCl 8.0, 200 mM NaCl, 2% glycerol, 0.5 mM Tcep, 800 nM Apro, 5 µg/ml Leu, 0.2 mM AEBSF, 0.05% DDM + 0.01% CHS) to obtain solution including protein. The protein solution concentrated to 2 ml was further purified by traversed SuperoseTM 6 10/300 GL with buffer (25 mM HEPES7.4, 200 mM NaCl, 0.5 mM Tcep 0.03% DDM + 0.006% CHS).

### Electrophysiological measurement assays

The experiment was implemented in a Vertical sample cell supplied by Warner Instruments, all current traces were recorded by a HEKA EPC 10 USB patch-clamp amplifier with a sampling frequency of 9900 Hz if not mentioned exclusively. One microliter of 25 mg/mL *E. coli* lipid which been extracted was precoated onto a 150 µM orifice of the cup, and 1 mL of the electrolyte solution (Trans-: 30 mM NaCl, 10 mM HEPES, pH = 7.5 ± 0.05; Cis-: 300 mM NaCl, 10 mM HEPES, pH = 7.5 ± 0.05) was added to both sides of the sample cell. Then, the lipid membrane was formed and the protein was embedded on the 150 µM orifice from the -cis side. After the protein was embedded, the solution was changed to 1 mL of solution without proteins. Then the successive experiment was performed. To verify the channel activity of Csg2, electrolyte solution (Trans-: 30 mM $CaCl_2$, 10 mM HEPES, pH = 7.5 ± 0.05; Cis-: 300 mM $CaCl_2$, 10 mM HEPES, pH = 7.5 ± 0.05) was used and 5 µL Csg2 protein extract with concentration of 1.2 mg/mL was added into the cis-chamber. When the applied voltage is applied, Csg2 is embedded within 20−30 min. The obvious current transition phenomenon can be observed, indicating that Csg2 is successfully embedded into the phospholipid membrane to form a channel. To evaluate the ion selectivity of Csg2, another ion conductance assay with asymmetric electrolyte solution (Trans-: 30 mM NaCl, 10 mM HEPES, pH = 7.5 ± 0.05; Cis-: 300 mM NaCl, 10 mM HEPES, pH = 7.5 ± 0.05) was conducted. After the Csg2 protein was embedded and stabilized for a period, the current value was adjusted to 0 by adjusting the applied voltage, then $Ca^{2+}$, $Mg^{2+}$, $Zn^{2+}$, $Mn^{2+}$, and $Fe^{2+}$ (the final concentration is 20 mM) were added successively into the *cis-* chamber. Record the influence value of the addition of ions on the current. After each ion

added, the voltage is adjusted to 0 when Csg2 stabilized, and subsequent ions are continued to be added for assay. In each independent replicate, set $\Delta ICa^{2+} = 1$, record $\Delta IX^{2+}$, and use $\frac{\Delta IX^{2+}}{\Delta ICa^{2+}}$ to characterize the relative anion permeability of each ion with respect to calcium ions. Although this in vitro substrate transport assay which performed with single-channel recording, did not replicate the exact physiological conditions of Csg2, it provided evidence that Csg2 had a preferred translocation selectivity for Calcium ion. In order to explore the effect of calcium ion on Csg2 channel, the current trace of 0/10/100/500 mM calcium ion concentration was recorded, and the single-channel power spectrum analysis was performed for each current trace.

### Proteinase K protection assay
200-OD yeast cells collected in 1.5 mL centrifuge tube were subsequently suspended in lysis buffer (50 mM Tris, 150 mM NaCl, 10% glycerol) with silica beads by tissue breaker to crush. The cells were homogenized and then centrifuged at $3000 \times g$ for 10 min to remove the nucleus. The supernatant was considered as post-nuclear supernatant (PNS). PNS were prepared from yeast cells expressing [85-]GFP[-86]-Csg2. PNS was incubated in concentration gradients of proteinase K for 1 h with or without 1% of Triton X-100 (V/V). TritonX-100 was used to dissolve the membrane for thorough digestion of proteins. Samples were prepared for immunoblotting analysis.

### Real-time qPCR with reverse transcription
Specially induced yeast cells were collected and then resuspended with 1 mL ZYMOLYASE-20T (2 mg/mL) (MP Biomedicals, 320921) that is an enzyme preparation from a submerged culture of Arthrobacter luteus to lyses cell wall at 30 °C for 1 h. Total RNA was prepared using the RNA isolation kit (Vazyme, RC101-01). A total of 1 μg RNA was reverse-transcribed using a Strand cDNA Synthesis Kit (Vazyme, R 323-01). Real-time PCR was conducted using ChamQ Universal SYBR qPCR Master Mix (Vazyme, Q711-02) on an iCycler RT-PCR Detection System (Bio-Rad Laboratories). Quantitation of all target gene expression was normalized to the control gene PGK1 for yeast genes. All of the data were obtained for at least three repeated experiments.

### Exogenous adding phytosphingosine (PHS) assays
Yeast cells were cultured in YPD medium to log phase, and then switched to SD-N medium and added phytosphingosine (STANDARDS, ZL-159084) dissolved in absolute ethanol in the meantime for indicated times at 30 °C. Cells were collected at specific times for Immunoblotting analyses, cell viability assays or fluorescence microscopy observation assays.

### Measurement of endoplasmic reticulum and total intracellular Ca²⁺
To measure ER calcium concentration with Cell ER Calcium Concentration Fluorescence Detection Kit (Shanghai HALING Biological Technology HL10267.1), which determines by Mag-Fluo-AM that is a low calcium affinity fluorescent labeling dye and is specifically captured by the endoplasmic reticulum and can detect the change of free calcium concentration in the endoplasmic reticulum to transform. Yeast cells were cultured in YPD medium to log phase, and then switched to SD-N medium for indicated hours. 1 mL yeast cells (OD600 = 1) were collected at 500 g/3 min in the 1.5 mL centrifuge tube, then followed the instructions. The relative fluorescence unit (RFU) was quantified using a fluorescence spectrophotometer with an excitation wavelength of 490 nm and an emission wavelength of 525 nm. The ER calcium concentration was calculated according to the formula [(Sample RFU-Blank control RFU)/ (Maximum control RFU-Sample RFU)] *22(μmol/L; Kd). All procedures were performed on ice.

Genetic endoplasmic reticulum Ca²⁺ fluorescent probe Stt3-jGCaMP7c was used for fluorescence observation. Stt3-jGCaMP7c and ER marker Cherry-HDEL were co-expressed in yeast cells and observed by fluorescence assays in after starvation in SD-N medium for indicated hours. Fluorescence parameters were set the same to compare the signal intensities between indicated cells. Experiments were conducted three times with independent clones of each yeast strain, each time capturing five different regions containing at least 100 cells in total. Representative images were shown.

To measure the total intracellular free Ca²⁺, Fungus/Yeast Calcium Ion Concentration Colorimetric Quantitative Detection Kit (Shanghai HALING Biological Technology, HL50097.7) was used, which determine by quantitative colorimetry via Ca²⁺ reacting with o-phenolphthalein complex ketone under alkaline conditions to produce a violet-blue complex. O-cresolphthalein complexone (CPC) colorimetric method is a stable, simple and accurate method for calcium detection. 100 mL yeast cells (OD600 = 1) after starvation in SD-N medium for indicated hours were collected at 500 g/3 min in the 1.5 mL centrifuge tube, then operated from the protocol. Absorbance readings at wavelength 570 nm were obtained using spectrophotometer. Constructed the standard curve and the calcium concentration (mmol/L) was calculated from it. All procedures were performed on ice.

### Co-immunoprecipitation
200-OD yeast cells collected in 1.5 mL centrifuge tube were crushed in lysis buffer (50 mM Tris, 150 mM NaCl, 10% glycerol) with silica beads by tissue breaker to get total proteins. Both EDTA-free protease inhibitor complete cocktail (Roche) and 20 mM N-Ethylmaleimide were added to lysis buffer to prevent from degradation and loss of proteins. Then, the supernatant containing total proteins collected by centrifugation was mixed with GFP-Trap beads (Chromotek) for 4 h with rotation at 4 °C. After centrifugation, beads were followed by a stringent washing procedure with wash buffer (lysis buffer with 1% NP-40) to remove nonspecific background binding, then which were cleaved with HU lysis buffer at 65 °C for 20 min and analyzed by immunoblotting.

### Lipidomics analysis
Indicated yeast cells were cultured in YPD medium to log phase, and then collected 100OD cells to 1.5 mL centrifuge tube or switched to lacking nitrogen medium with treatment for indicated hours before collection. The collected cells were immediately snap-frozen in liquid nitrogen and transported on dry ice for lipidomic analysis.

Lipidomics analysis of yeast cells was conducted at LipidALL Technologies using an MRM library constructed based on the company's preceding publications on yeast lipidomes[145–147]. Lipids were extracted from yeast cells using standard Bligh and Dyer's method as described previously[111]. Briefly, yeast cells were homogenized in 750 μL of chloroform:methanol 1:2 (v/v) containing 10% deionized water with glass beads on an automated bead shaker (OMNI, USA). The homogenate was then incubated at 800 g for 1 h at 4 °C. At the end of the incubation, 350 μL of deionized water and 250 μL of chloroform were added to induce phase separation. The samples were then centrifuged and the lower organic phase containing lipids was extracted into a clean tube. Lipid extraction was repeated once by adding 500 μL of chloroform to the remaining tissues in aqueous phase, and the lipid extracts were pooled into a single tube and dried in the SpeedVac under OH mode. Samples were stored at −80 °C until further analysis.

Lipidomics methodology was reported according to standard guidelines[113]. Polar lipids including the classes of PE, PA, PI, PS, PG, LPE, LPA, LPI, LPS, and FFAs were analyzed in the negative ion mode under electrospray ionization (ESI) on an Agilent 1260 HPLC coupled to Sciex 5500 QTRAP, with source parameters as follows, CUR 10, TEM 400 °C, GS1 20, GS2 20. Other polar lipid classes such as PC, LPC, Sph, and PhytoCer were analyzed in the ESI positive ion mode on the same machine, with source parameters CUR 10, TEM 400 °C, GS1 30, GS2 30. Separation of individual lipid class of polar lipids was carried out using a Phenomenex Luna 3 μm-silica column (internal diameter

150 × 2.0 mm) with the following conditions: mobile phase A (chloroform:methanol:ammonium hydroxide, 89.5:10:0.5) and mobile phase B (chloroform:methanol:ammonium hydroxide:water, 55:39:0.5:5.5). The gradient started with 2% B, which was maintained for 1 min before increasing to 35% B over the next min. %B was further increased to 65% over the next 5 min, and finally reached 100% B at the 8th min. The gradient was maintained at 100% B for 5 min, and then decreased back to 2% B and equilibrated for 4 min prior to the next injection. Flow rate was 350 μL/min and column oven temperature was at 35 °C. Individual lipid species were quantified by referencing to spiked internal standards, which included d9-PC32:0(16:0/16:0),d7-PE33:1(15:0/18:1),d31-PS,d7-PG33:1(15:0/18:1),d7-PI33:1(15:0/18:1),d7-PA33:1(15:0/18:1),Cer(d18:1-d7/15:0),d7-LPC18:1,d7-LPE18:1,C17:0-LPA,C17:1-LPI,C17:1-LPS,d17:1-Sph from Avanti Polar Lipids. Inc. Free fatty acids were quantitated using d31-16:0 (Sigma-Aldrich) and d8-20:4 (Cayman Chemicals) as internal standards. Glycerol lipids including diacylglycerols (DAG) and triacylglycerols (TAG) were quantified using a modified version of reverse phase HPLC/MRM on an Agilent 1260 coupled to SCIEX QTRAP 5500 under ESI positive ion mode[153]. Separation of neutral lipids were achieved on a Phenomenex Kinetex-C18 2.6 μm column (i.d. 4.6 × 100 mm) using an isocratic mobile phase containing chloroform:methanol:0.1 M ammonium acetate 100:100:4 (v/v/v) at a flow rate of 170 μL for 17 min. Relative quantities of TAGs were calculated by referencing to spiked internal standard TAG (16:0)3-d5 obtained from CDN isotopes, while DAGs were quantified using d5-DAG18:1/18:1 from Avanti Polar Lipids. Free cholesterols and cholesteryl esters were analyzed under atmospheric pressure chemical ionization mode in the positive polarity on an Agilent 1260 HPLC coupled to SCIEX QTRAP 5500 as described previously, with d6-cholesterol and d6-C18:0-cholesteryl ester (CE) (CDN isotopes) as internal standards[152].

### Statistical analysis

All experiments were independently repeated at least three times with consistent conclusions and representative results were shown. The values are expressed as means ± SD at least three independent experiments. The significance analysis between different groups was determined by two-way ANOVA tests of variance using the GraphPad Prism software (version 8.0). $p < 0.05$ was considered statistically significant, $p > 0.05$ was considered statistically nonsignificant.

### Reporting summary

Further information on research design is available in the Nature Portfolio Reporting Summary linked to this article.

## Data availability

The data that support this study are available within the paper and in the Supplementary Information files and from the corresponding authors upon request. Source data are provided with this paper.

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

## Acknowledgements

We thank Dr. Ivan Psakhye and Dr. Fabian den Brave for early discussions, Jochen Rech and Alexander Strasser for technical assistance, Dr. Philip Hieter, Dr. Peter C. Stirling, Dr. Zhiping Xie and Dr. Cong Yi for yeast strains and plasmids. This study was supported by the National Natural Science Foundation of China under grants 31970693 (to K.L.) and 32022020 (to K.L.) and 81902997 (to H.L.), the Disciplinary Excellence Development 135 program of West China Hospital under grant ZYYC20015 (to K.L.) and the Sichuan Province Science and Technology Project under grant 2020JDJQ0015 (to K.L.).

## Author contributions

S.L., Y.L., Y.Z., and T.H. conducted the experiments in yeast cells; M.C. and J.G. performed the in vitro ion channel experiments; Y.W. and S.Q. purified the Csg2 proteins; H.L. and S.M.L. helped on analyzing the lipidomics data; K.L. designed the project and coordinated the experiments and analyzed the results; K.L. wrote the paper.

## Competing interests

The authors declare no competing interests.
