## [Peer Review File · Nature Communications]

The ER Calcium Channel Csg2 Integrates Sphingolipid Metabolism with AutophagyREVIEWER COMMENTS

Reviewer #1 (Remarks to the Author):

In the manuscript entitled „The ER Calcium Channel Csg2 Integrates Sphingolipid Metabolism with Autophagy“ by Liu et al. the impact of the calcium channel csg2 on autophagy in yeast is investigated. The authors performed and describe a huge panel of experiments to unravel the role of sphingolipid metabolites on autophagy. By employing various deletion mutants and screens they decipher the role of calcium in controlling the activity of enzymes involved in the complex sphingolipid pathway.

All in all the presented data are well documented and very interesting, linking the calcium levels in the ER to sphingolipid metabolism and the control of autophagy.

The introductory part as well as the discussion section should give more precise informations. For example, in lines 38-42 the authors write that a steep concentration gradient exists across the plasma membrane and vacuolar membrane (approx. 10.000 fold). This should be checked and correct numbers should be given for the subcellular compartments and also for prevailing calcium levels in the extracellular space.

I.52: what do the authors mean by “modulating the driving force of Ca transfer “?

The next sentence (l. 54 – 65) states that „Calcium channels indirectly regulate a variety of calcium-related processes, including cell proliferation, apoptosis, and autophagy“. Specific examples should be given here to show the impact of calcium in these metabolic pathways.

I. 56 – 59: Same as above. The introduction would benefit from more precise details. Here, it would be helpful to list/describe examples of metabolic consequences elicited by abnormal calcium accumulation in subcellular compartments and to refer to data, which delineate the consequences of dysfunctional calcium channels.

I.74 – 77: This controversy need to be discussed in the section „Discussion“.

According to the data presented in Fig. 1b, the authors focused on csg2 due to the limited starvation resistance after 72h SD-N medium supply. The authors should also comment on the data shown for pmr1Δ. It is unclear to me how to explain the reduced survival rate after 3h and 16h but a higher survival rate after 72h - while autophagy is not affected after 16h in pmr1Δ cells (see Fig. 1C)

I. 124: The experimental setting (here Fig. 1D) needs to be described, then the obtained data and the conclusions drawn. This is a general comment to improve the presentation of the data and readability of this manuscript.

The authors overinterpret their results in the paragraph starting with line 190. Here the authors refer to the structural domains of csg2, which were investigated with respect to their necessity to control autophagy. The authors state that all ten TM domains are required for the function. This might be so but is not shown by the experiments since no individual – or better - pairwise deletion of the TM regions

were performed. The overinterpretation should be avoided but the necessity of the TM domains (the biological significance) for functioning of the channel should be briefly discussed.

The EF-hand structure is described to contain 40 residues (line 194). What is meant by „residues“? The amino acid range is given from 95 – 107 (Fig S2F), which is 12 amino acids. This should be clarified. It would further be of interest to know whether the Ca-binding EF hand is located in the ER lumen or cytosolically. It should be considered (and discussed) that the Ca-channel Csg2 is regulated by calcium ions.

I. 146: The authors state that csg2 regulates autophagosome fusion. To my mind a channel does not regulate autophagosome fusion. It rather “is involved in”.

I. 207/208: The conclusion drawn here do not really reflect the data and results shown in this paragraph, since the question was whether TM and or EF hand motifs are required for csg function.

I. 247 -252: starting with „We speculate that as the calcium probe jGCaMP7c“. First the result should be described and then it should be interpreted. The reasoning given here is not easy to understand. Isn't it more likely that the overall content of calcium in the cells is comparable but that the distribution within subcellular compartments is different?

Could the authors please comment on the experimental data in Fig S3E? In csg2Δ cells, treated with EDTA, the amount of free calcium is primarily reduced in the cytosol, not in the ER. How does it look like after 16h starvation in SD-N medium (which is the experimental setting for the subsequent studies)? It would be also of interest to compare the calcium levels (in Fig S3E) with the calcium levels in the different subcellular compartments in WT yeast cells.

There is a high amount of information, especially in Fig. 6 and Fig. S6. I would appreciate a more careful description of the experiments, the results and the interpretation – possibly with an additional illustration, in which the results are combined within one scheme. This would largely facilitate the reading of the manuscript and the deciphering of the meaning.

The metabolic link between PHs and Aur1 activity is not clearly presented.

I. 410/423: According to the data it should be discussed that Aur1 stability is controlled by prevailing calcium concentrations. Another thesis than the presented ones is the possibly that Aur1 protein forms aggregates in the presence of high calcium levels (aggregate are shown in Fig. 7C) and is subsequently degraded leading to the determined low protein levels of Aur1.

Having analysed the data shown in Fig.3 and Fig.S3 I have the impression that it would be of interest to determine the sensibility of sur2 activity to various calcium concentrations. Possibly, high calcium concentrations activate sur2 resulting in high amounts of PHS and inhibition of autophagy. Vice versa, low amounts inhibit sur2 activity – reducing PHS amounts, which enables autophagy. The addition of PHS again blocks autophagy. Could the authors comment on this hypothesis?

In the Discussion I. 446-448: To my mind here the authors contradict the data shown in Fig 5C'.

In Fig. 3. The authors state in the legend D) “transportation of RFP-Ape1 was observed In atg1Δ cells“. It was investigated but not observed !

Due to language problems some of the paragraphs/sentences are difficult to understand, sometimes the meaning remains obscure.

Legend to Fig. 4A: should be rephrased

In general, there are language problems and at some passages it is almost not possible to catch the meaning. Editing by a native speaker is required.

Just to list some examples:

Sentence l. 50 -54: „... by enabling calcium influx into and out of the environment,...“

l. 128: should be rephrased, for example: “The degradation of GFP-50Q, another..., was shown to be blocked in *csg*-depleted yeast cells”.

l. 199: "... showed no adverse effect on ...”

rephrasing also required l. 214, l. 224, l. 239 – 240, l. 242, l. 247-252, l. 302/303, l. 345-350, l. 394-397, l. 408: the term “necessary” should be “physiological” or “adequate”.

Reviewer #2 (Remarks to the Author):

Calcium levels in the cell are critical for the regulation of many signaling pathways. However, despite the critical role of calcium in signaling in cells it is not clear if calcium levels regulate autophagy. In the manuscript by Liu et al the authors investigate the potential link between cellular calcium levels and autophagy by investigating whether any ER resident calcium channels are required for autophagy. They initially tested this by monitoring the survival of yeast strains containing knockouts of different ER resident calcium channels in media lacking nitrogen sources (SD-N). They demonstrate that *csg2Δ* yeast show a significant loss of survival in SD-N suggesting that autophagy may be blocked in these yeast. They utilize various autophagy assays to demonstrate that *csg2Δ* yeast cannot undergo autophagy in response to nitrogen starvation. They next demonstrate that *csg2* is indeed an ER resident calcium channel using cellular localization and calcium transfer assays and that *csg2Δ* yeast have increased levels of calcium in the ER. They show that the addition of EDTA or EGTA to chelate calcium leads to a partial restoration of autophagy in *csg2Δ* cells in SD-N media and that deletion of *pmr1* (an ATPase that is required for calcium uptake by the ER) leads to a partial restoration of autophagy. Next, they demonstrate that deletion of *csg2* leads to the accumulation of sphingolipids and that a deletion of *sur2* (required for sphingosine production) leads to a recovery of autophagy in *csg2Δ* yeast suggesting that the accumulation of sphingosine results in autophagy inhibition. Lastly, the authors demonstrate that *csg2* deletion leads to a reduction in the amount of *aur1* (required for complex ceramide production) in the cell. Taking everything together the authors propose a model in which deletion of *csg2* from yeast leads to the accumulation of calcium in the ER and decreased *aur1* resulting in increased sphingosine and blocked autophagy.

This is a very thorough investigation into the topic of calcium accumulation and autophagy. One major strength of this manuscript is that the authors are careful to use multiple assays to validate individual observations. This increases the overall confidence in the data presented. However, one major weakness of the manuscript is that the authors are unable to block autophagy in WT cells based on the model they propose. This makes it unclear how calcium and SPH may regulate autophagy in WT cells. The authors do attempt to block autophagy in WT cells by adding calcium and PHS which based on the model would be predicted to block autophagy. However, this is not what the authors observed. This suggests that there might be a missing piece or two in their model or that calcium accumulation in the ER in WT cells does not block autophagy. I do acknowledge that the authors show that PHS is able to block autophagy in *csg2Δsur2Δ* cells but I think a key experiment would be one in which the authors are able to recapitulate the block of autophagy in WT cells based on their model. While this would be a helpful validation of their model I don't believe it is critical for the final manuscript. However, it would be helpful for the authors to acknowledge this caveat and state that a piece of the puzzle model is missing at this time since it is unclear how calcium accumulation may regulate autophagy in WT cells.

Major concerns

- 1) The authors demonstrate that the deletion of *csg2Δ* leads to a block of autophagy in response to nitrogen starvation. Do the authors believe that this block of autophagy only occurs in response to starvation (nitrogen or glucose)? There are several forms of basal autophagy that go on during times when cells have access to nutrients including the Cvt form of selective autophagy. Therefore, it would be informative for the authors to investigate Ape1 processing when nutrients are plentiful to determine if autophagy is blocked even prior to nutrient starvation.
- 2) The authors do not adequately explain the Electrophysiology Measurement Assay or the Cell ER Calcium Concentration Fluorescence Detection Kit assay in enough detail. The authors should include more details for these assays in the methods. In particular, the authors should provide more details as to what is meant by "after the protein was embedded 1ml of solution was changed ..." on line 565 and 566. Is the Cell ER Calcium Concentration Fluorescence Detection Kit assay a microscopy-based assay? If so, the authors should include representative microscopy images.
- 3) Based on the model proposed by the authors one would predict that adding calcium to yeast cells should lead to an inhibition of autophagy in WT cells in SD-N. However, in Figure 3A this is not what the authors observe, and the authors do not adequately acknowledge this. Do the authors have any speculation as to why they do not observe autophagy inhibition when calcium is added to WT cells? In addition, given the results from this figure the authors may want to consider changing the title of the figure as adding calcium did not lead to a blockage of autophagy in WT cells. It may also be helpful to monitor calcium accumulation by *stt3-jGCaMp7c* in WT cells in the absence and presence of added calcium as this may show if there is an increase in ER calcium when CaCl_2 is added and may help explain their results.
- 4) In a similar comment to point 3. When the authors add PHS to WT cells in figure 6 it does not lead to a blockage of autophagy. It seems that given the model this would be predicted to be the case. Do the authors have any speculation on why this might be the case? The authors should acknowledge this

result in the manuscript and clearly describe in their model that they do not observe autophagy inhibition in WT cells when calcium or PHS is added.

5) In Figure 6F the authors knockout Lcb4 and Lcb5 to test the role of different enzymes on autophagy inhibition. Would the authors expect that a deletion of Lcb4 and Lcb5 should lead to an accumulation of PHS? Since Lcb4 and Lcb5 use PHS as a substrate one would predict that this might be the case. If Lcb4 and Lcb5 knockout does lead to an increase in PHS, why does deletion of Lcb4 and Lcb5 in WT cells not lead to a block of autophagy?

Reviewer #3 (Remarks to the Author):

The manuscript by Liu et al., implicates that calcium homeostasis in the endoplasmic reticulum (ER) maintained by calcium channel Csg2 is involved in regulation of autophagy. Csg2, which is essential for yeast growth in the presence of high calcium concentrations and a potential channel for calcium (Beeler et al., 1994), has previously been shown to function in mannosylinositol phosphorylceramide (MIPC) synthesis as a regulatory subunit in a complex with Csg1 or Csh1, although the precise role in that reaction has not been defined (Umemura et al., 2003). In this paper from Liu et al., the authors give evidence that Csg2 is an ER resident calcium channel mediating calcium release. In addition, the authors show that starvation-induced autophagy is blocked in Csg2-deleted yeast (*csg2Δ*) cells and the blockade is caused by increased calcium levels in the ER. Since a comprehensive lipidomics analysis exhibited that phytosphingosine (PHS) and phytoceramides, precursors of complex sphingolipids such as inositolphosphorylceramide (IPC) and MIPC, accumulate in *csg2Δ* cells after starvation, the authors hypothesized that the increase in PHS level causes blockage of autophagy in *csg2Δ* cells. In addition, it was shown that mutations of Lip1 (subunit of ceramide synthase) and Aur1 (subunit of IPC synthase) suppress vacuolar degradation of Atg8-GFP upon nitrogen starvation and Tsc10 (which catalyzes the second step in PHS synthesis) mutation restored the defect of Atg8-GFP degradation in *csg2Δ* cells. Furthermore, the manuscript showed that Aur1 protein level is reduced in *csg2Δ* cells. Based on these data, the authors argue that the increase in ER calcium levels caused by Csg2 deletion is responsible for reduced Aur1 level, leading to increased PHS level that inhibit autophagy.

The experiments have been carried out well and were reported clearly. There is information in the work that will be of interest to the field of autophagy. The manuscript presents strong evidence for an important role of ER calcium channels in regulation of autophagy. Unfortunately, however, it has several serious flaws. The major one is the conclusion that PHS regulates autophagy, which is speculative because it is based solely on an analysis of GFP-Atg8 processing and starvation resistance. It has not been determined which steps are inhibited by PHS and it remains possible that PHS affects fusion with the vacuole, which is downstream of the step involving calcium, or vacuole acidification which is necessary for protein degradation. Important experiments to validate this point and the data are missing. It is also necessary to add control data confirming that mutations of Lip1 and Aur1 accumulate PHS, while mutation of Tsc10 decreases PHS level in WT and *csg2 Δ* cells. Moreover, as exogenously

added PHS may have an artificial effect, in addition to experiments with PHS addition, further analysis should be conducted to rule out the possibility that blockage of autophagy in *csd2Δ* cells is caused by reduced PHS-type MIPC level (reduced MIPC levels in *csd2Δ* cells are not shown in Figure 5, and it should be shown). It is also unclear how *Csg2* deletion (accumulated ER calcium) reduces the protein level of *Aur1* and why *Aur1* and *Kei1*, which are known to localize to the Golgi apparatus (Levine et al., 2000; Lisman et al., 2004; Sato et al., 2009), are localized to the ER in this study. There remain the substantial issues with the study that would need to be addressed to strengthen the claims before the manuscript would be suitable for the Nature Communications. As such, the work is premature, but might be improved by the above comments and the following further experiments.

Major comments ;

1. As mentioned above, it remains possible that PHS affects fusion with the vacuole or vacuolar acidification. It should be determined which steps are inhibited by PHS.
 2. Based on the results with *tsc10*, *lip1*, and *aur1* mutant strains and double mutants with *csg2 Δ*, the authors claim that accumulated PHS causes blockage of autophagy after starvation. However, the manuscript does not provide any experimental evidence to confirm the levels of PHS and other sphingolipids. Lipid analysis needs to be done as this is directly relevant to the conclusion.
 3. Since *Csg2* has been shown to play an essential role in MIPC synthesis, the manuscript should provide strong experimental evidence to eliminate the possibility that reduced MIPC levels in *csd2Δ* cells cause blockage of autophagy. Further analysis needs to be done. For example, *rsb1* deletion mutant (Kihara and Igarashi Y, 2002) or *ypc1* mutant (Mao et al., 2000) strain combined with *lcb4* and *lcb5* mutations that does not affect MIPC synthesis but is expected to increase PHS levels might be used for the analysis. In addition, if the authors can show that the autophagy defect are restored in *csg2Δ* background with normal levels of IPC but reduced levels of PHS, then the possibility that increased IPC levels in *csg2Δ* cells cause blockage of autophagy would be also ruled out, although it may be difficult to create such lipid level conditions.
- Csg1* and *Csh1* function in MIPC synthesis as catalytic subunits (Uemura et al., 2003). Although *csg1 Δcsh1Δ* double mutant exhibits a calcium sensitivity, this sensitivity has been proposed to be caused by an accumulation of IPC-C, one of IPC classes. If ER calcium accumulation in *csg2Δ* cells directly causes reduced level of *Aur1* protein and increases the level of PHS, *csg1 Δcsh1Δ* double mutant may not affect *Aur1* stability. If the authors can show a specific role of *Csg2* and that *Csg1* and *Csh1* are not involved in autophagy regulation, the data will be more convincing evidence to support the proposal.

4. Does the decrease in Aur1 protein level in the *csg2* Δ strain occur only during starvation? What about in nutrient rich media? Is it degraded by the ubiquitin (Ub)-proteasome pathway or in vacuoles? Do the protein levels of Aur1 in the *csg2 pmr1* double mutant strain remain normal?

5. Aur1 and Kei1 has been well known to localize to the Golgi apparatus (Levine et al., 2000; Lisman et al., 2004; Sato et al., 2009). Is the ER localization of Aur1 and Kei1 shown in this study a starvation-specific phenomenon? Aur1 and Kei1 localization in nutrient rich media should be shown. It is also necessary to analyze whether the dot-like compartments of Aur1-GFP in the *csg2* Δ cells are Golgi apparatus.

6. If Aur1 relocates from the Golgi to the ER during starvation, sphingolipid metabolism will be severely affected by starvation. Figure 5 shows the difference in lipid levels between WT and *csg2* Δ strains after starvation and the data before starvation are missing. The differences in lipid levels before and after starvation, as well as between WT and *csg2* Δ strains before starvation should be analyzed to determine if the increase in PHS and ceramide levels in *csg2* Δ cells is specific to starvation. A sentence in ABSTRACT "Deletion of Csg2 causes increases in calcium levels in the ER and then disturbs Aur1 stability, leading to accumulation of the bioactive sphingolipid phytosphingosine....." will mislead the reader. Therefore, it needs to be clarified whether changes in lipid levels, as well as Aur1 stability and localization are specific to starvation or not.

7. The authors claim that accumulated PHS causes blockage of autophagy after starvation. Explain why the addition of PHS to WT cells does not inhibit autophagy (Figure 6J and L).

8. Exogenously added PHS is linked to de novo synthesis pathway via phosphorylation by Lcb4/Lcb5 and dephosphorylation by Lcb3 processes (Funato et al., 2003). Therefore, exogenously added PHS is expected to not block autophagy in *csg2* Δ *sur2* Δ *lcb3* Δ or *csg2* Δ *pmr1* Δ *lcb3* Δ (or not in *lcb3* Δ single mutant if the addition of PHS to WT cells inhibits autophagy). To confirm the importance of PHS in autophagy regulation, the effects of *lcb3* deletion should be shown.

9. It was not tested whether the function of Aur1 is disrupted by increase in PHS. Does PHS addition affect Aur1 protein level and localization in *csg2* Δ *sur2* Δ or *csg2* Δ *pmr1* Δ , as well as in WT cells?

10. The authors argue that calcium accumulation causes blockage of autophagy through dysfunction of Aur1. If so, Aur1 overexpression or artificial localization (for example, utilizing GFP-GFP-binding protein (GBP)) may suppress blockage of autophagy. It can be tested.

Minor comments ;

1. In Fig S3E, is dot-like compartments of ER calcium probe Stt3-jGCaMP7c in *csg2Δ* with CaCl₂ the ER? DIC images of cells with CaCl₂ are different from others. Is cell death beginning to occur?

2. In Fig S6B, the enzyme that converts PHCer to PHS is Ypc1, not Ydc1.

3. In the third paragraph of the Discussion, the author states that "although we established the link between PHS and autophagy inhibition at the autophagosome formation stage, how this inhibition is achieved is unclear.". However, as noted in major comments 1, the authors have not examined which steps are inhibited by PHS. It should be expressed accurately.

Reviewer #4 (Remarks to the Author):

In this study, Liu et al identify the yeast calcium channel/MIPC synthase subunit as an important effector of autophagy. Deletion of *csg2* results in altered calcium homeostasis of the ER and almost completely abolishes the formation of autophagosomes upon starvation, either nitrogen – or glucose starvation. The in vitro data on calcium transport are very nice. The authors go on to show that the levels and localization of the IPC synthase Aur1 are altered and use lipidomics techniques to identify changes in the lipid composition of *csg2Δ* cells. Through genetic analyses of different temperature sensitive mutants the authors show that changes in SL composition, and especially the levels of phytosphingosine, are responsible for the block of autophagy in CSG2 mutant cells.

Most experiments have been carried out well and have the appropriate controls. The manuscript is also written well and clearly. Unfortunately, the manuscript has, in my opinion, some serious flaws that have to be addressed. The lipidomics data have to be presented differently and have to be better annotated. Many of the observations in the manuscript have been, in very similar form, made before. On the other hand, several important previous studies and findings are not really addressed in the manuscript or at least in the discussion. An example would be the role of Csg2 in the stabilization of the MIPC synthase subunits Scg1 and Csh1. Also the fact that Tsc10 is, in fact, identified as a "Temperature-sensitive Suppressors of Csg2 mutants" (hence its name) is not mentioned in the manuscript. In addition, while several observations are very interesting, the overall connection between calcium homeostasis, sphingolipid metabolism and autophagy remains largely unexplored. There remain substantial issues that have to be addressed to strengthen the manuscript. Maybe the following suggestion can help the authors to improve the manuscript.

Major points:

- The lipidomics data have to be completely revised. From the material and methods section it is not clear what was measured, which quality controls were used and how the lipid species shown were selected. To the best of my knowledge yeast has only saturated and mono-unsaturated fatty acids. Why are lipid species such as PC 36:5 shown in the analysis. Why is cholesterol part of the analysis when yeast only produces ergosterol as the major sterol. In which ion mode were the data obtained. Some of the species are mostly detectable in positive ion mode (TAGs, DAGs, PC) other are better detected in negative ion mode (PI, PS, PG). Why were the lipids presented chosen. In the results table there needs to be an annotation of the Q1 and Q3 masses used to determine the shown lipid species. In addition, the retention times of the standards and the chosen lipid species need to be presented. I would recommend to include the checklist from a recently published paper (doi: 10.1038/s42255-022-00628-3) here. Since this was also published by the nature publishing group this should be incorporated here.
- The major effect on sphingolipid levels resulting from a CSG2 deletion is the depletion of MIPC and M(IP)2C with a concomitant increase in IPC levels. Complex sphingolipid should be included in the lipidomics analysis. CSG2 mutants show a two-fold increase in PHS. According to the authors this should be sufficient to block autophagy. What are PHS levels in the other mutants analyzed? Does the inhibition of the ceramide synthase with fumonisins B1 block autophagy?
- The authors speculate in the discussion that increased PHS levels inhibit the TORC1 complex to block the formation of autophagosomes. This would be an important point to strengthen the manuscript. Assays to determine TORC1 activity in yeast are available and this should be tested. How do the authors explain the accumulation of PHS in the ER to affect the vacuolar TORC1 complex?
- Previous manuscripts show the effect of CSG2 deletion on stability of the MIPC synthase. This is completely ignored in the manuscript and especially the discussion.

-

Minor points:

- According to the figure legends, temperature sensitive mutants were grown at non-permissive temperature for 16 hours. How many cells are still alive after such a long exposure to the non-permissive temperature?
- The authors should be careful regarding their lipid annotation. For example, in figure 7a in the upper panel, PHS is used. In the other panels the authors use "sphingosine". Sphingosine has a double bond and does not exist as a molecule in yeast cells. Only PHS and DHS are present and according to the results of the SUR2 mutants it makes a big difference for the outcome. Thus, the nomenclature should be treated very carefully.
- The model in figure 1a is also inaccurate. Why does the ER have such a big lumen. I assume that is supposed to be the nucleus. This should be visualized with more care.
- Can the authors include a control that shows the functionality of GFP tagged Csg2, eg. in autophagy (pho8D60 assay) or starvation resistance?

While most of the data is presented nicely the mechanistic connection between calcium homeostasis, sphingolipid biosynthesis and autophagy is not clear from this manuscript. In its current form I cannot recommend this manuscript for publication in Nature Communications.

Point-by-point responses to the reviewers

All the authors sincerely thank the editor and the reviewers for their time and effort in handling our manuscript "The ER Calcium Channel Csg2 Integrates Sphingolipid Metabolism with Autophagy". We thank the appreciation of our work from the reviewers and the very helpful suggestions put forth by the reviewers, which have greatly improved the quality of this work. We have substantially revised the manuscript and addressed all issues raised by the reviewers, as shown point-by-point below.

Reviewer #1

We thank the reviewer for the time and the positive comments "*All in all the presented data are well documented and very interesting, linking the calcium levels in the ER to shingolipid metabolism and the control of autophagy.*".

The specific comments are addressed as shown below.

Specific comments:

1) "*The introductory part as well as the discussion section should give more precise informations. For example, in lines 38-42 the authors write that a steep concentration gradient exists across the plasma membrane and vacuolar membrane (approx. 10.000 fold). This should be checked and correct numbers should be given for the subcellular compartments and also for prevailing calcium levels in the extracellular space.*"

Response: We thank the reviewer for this suggestion. More detail information is provided (line 39 - line 45): The concentration of calcium is maintained at very low levels in the cytosol (0.1 μM), and a steep calcium gradient exists across the plasma membrane (extracellular space calcium at 1 mM) and the membranes of intracellular calcium reservoirs (endosome calcium at 2 μM , acidic lysosome calcium at 0.5 mM, endoplasmic reticulum calcium at 0.5 mM, Golgi apparatus calcium at 0.1 mM, mitochondria calcium at 0.1 μM and secretory vesicle calcium at 0.1 mM)²⁻⁴.

2) *“l.52: what do the authors mean by “modulating the driving force of Ca transfer“?”*

Response: We are sorry for this obscure sentence. The “modulating the driving force of Ca transfer” was used to describe that calcium channels determine the directions of calcium transfer. These words are corrected and the new sentence is (line 53 - line 57): Calcium-permeable ion channels contribute to rapid changes in cytosolic calcium concentrations by enabling calcium influx into or out of the environment, and by determining the intracellular pathways for calcium uptake into or release from cellular organelles such as lysosomes/vacuoles and the ER^{7,12,13}.

3) *“The next sentence (l. 54 – 65) states that “Calcium channels indirectly regulate a variety of calcium-related processes, including cell proliferation, apoptosis, and autophagy”. Specific examples should be given here to show the impact of calcium in these metabolic pathways.”*

Response: We thank the reviewer for this suggestion and specific examples are given (line 59 - line 70): For example, increasing calcium activates calcineurin to dephosphorylate TORC2 and activate PKC and RasGRP to initiate the MAPK cascade, eventually activating genes that regulate proliferation¹⁴. Under stress conditions, the mitochondrial matrix is rapidly overloaded with calcium, which is the first step of apoptosis induction^{15,16}. The roles of calcium in autophagy are controversial. Studies have shown that calcium promotes autophagy by inhibiting the activity of mTOR through CAMKK2/CaMKKb (calcium/calmodulin dependent protein kinase kinase 2) and AMPK¹⁷⁻²⁰, and other studies have shown that an increase in intracellular calcium negatively regulates autophagy by promoting the binding of Ca²⁺-CALM/calmodulin to PIK3C3/VPS34 and subsequent mTOR activation^{21,22}.

4) *“l. 56 – 59: Same as above. The introduction would benefit from more precise details. Here, it would be helpful to list/describe examples of metabolic consequences elicited by abnormal calcium accumulation in subcellular compartments and to refer to data, which delineate the consequences of dysfunctional calcium channels.”*

Response: We thank the reviewer for this suggestion. As suggested, we added such background information (line 73 - line 88): Studies have shown that disruption of calcium channels in organelles such as the ER and mitochondria is correlated with abnormal insulin reactions and metabolic alterations in the liver²⁴⁻²⁶. Obesity-induced dysfunctional SERCA, an ER calcium channel, causes disrupted calcium homeostasis in the ER and impaired protein folding capacity²⁷. The importance of such a mechanism in the development of fatty liver and diabetes is further supported by studies showing that SERCA overexpression enhances the protein folding function of the ER, leading to upregulated glycemic control²⁸. Excessive calcium accumulation has been found to be associated with mitochondrial dysfunction^{29,30}. The expression of the calcium channel IP3R was increased significantly in the mitochondria-associated ER membranes (MAMs) of liver cells from different models of obesity³¹. Such alteration caused increased calcium flux from the ER to mitochondria, and this calcium overload resulted in increased mitochondrial reactive oxygen species (ROS) production and decreased oxidative phosphorylation^{29,30}. Consistently, suppression of IP3R restored mitochondrial function and significantly improved glucose tolerance in obese model mice³¹.

5) *“l.74 – 77: This controversy need to be discussed in the section „Discussion“.”*

Response: We thank the reviewer for this suggestion and this controversy is discussed in the section Discussion (line 549 - line 564): The mechanism by which Csg2-mediated calcium homeostasis regulates autophagy needs to be verified to determine whether it is conserved in mammalian cells. Of note, the roles of calcium in autophagy are rather controversial in mammalian cells, with many studies showing an inhibiting effect of calcium on autophagy, contrasting with the stimulating action of calcium shown in other studies⁵. Overall analysis of these studies showed that these controversies come from complex intracellular calcium signaling⁵. The eventual effects of calcium on autophagy greatly rely on spatiotemporal characteristics and the amplitude of calcium signals, the cell growth conditions, the tissue characteristics of cells, and the types of autophagy (basal or selective autophagy)⁵. Although it seems that

many parameters can determine the stimulatory or inhibitory effects of calcium on autophagy, calcium essentially regulates autophagy. As cytosolic and organellar calcium homeostasis is regulated by calcium channels, the illustration of correlations between calcium channels and autophagy activities under various situations will help discover the calcium-dependent mechanisms of autophagy regulation.

6) “According to the data presented in Fig. 1b, the authors focused on *csg2* due to the limited starvation resistance after 72h SD-N medium supply. The authors should also comment on the data shown for *pmr1Δ*. It is unclear to me how to explain the reduced survival rate after 3h and 16h but a higher survival rate after 72h - while autophagy is not affected after 16h in *pmr1Δ* cells (see Fig. 1C)”

Response: We speculated that autophagy is abnormally upregulated in *pmr1Δ* cells at the very beginning of starvation while autophagy in *pmr1Δ* cells is similar with WT cells after longer time of starvation. The abnormally early upregulated autophagy in *pmr1Δ* cells decreased cell survival rate compared with WT cells at the beginning of starvation (3h and 16h). However, after long time of starvation (72h), autophagy in both *pmr1Δ* cells and WT is induced to high levels, therefore, the survival rate of *pmr1Δ* cells is similar with that of WT cells. This speculation is supported by the experimental results (Fig. S1k). Upon short time of starvation, *pmr1Δ* cells but not WT cells showed obvious autophagic degradation of GFP-Atg8 (appearance of cleaved GFP moiety); after longer time of starvation, autophagic degradation of GFP-Atg8 in *pmr1Δ* cells is similar with that in WT cells (Fig. S1k).

In one previous study to screen for mitophagy modulators in yeast, it was also found that both non-selective autophagy and mitophagy were upregulated in *pmr1Δ* cells (Fig.3 in Reference PMID: 25704822 (Synthetic quantitative array technology identifies the Ubp3-Bre5 deubiquitinase complex as a negative regulator of mitophagy. Cell Rep. 2015 Feb 24;10(7):1215-25.))

We speculated that deletion *Pmr1*, the high-affinity P-type ATPase mediating calcium transport into the ER, causes low levels of calcium in ER, which therefore is beneficial for autophagy induction. The higher autophagic

activities in *pmr1* Δ cells is consistent with investigation that *csg2* Δ cells showed blocked autophagy resulting from accumulated high levels of calcium in ER.

7) *“ l. 124: The experimental setting (here Fig. 1D) needs to be described, then the obtained data and the conclusions drawn. This is a general comment to improve the presentation of the data and readability of this manuscript.”*

Response: We thank the reviewer for this suggestion. Now we describe the experimental setting, the obtained data and the conclusions drawn for results in the manuscript. Here for Fig. 1D (line 153 - line 157): Autophagic degradation of GFP-Atg8 upon starvation was monitored by detecting the cleaved GFP moieties, and the results showed that in *csg2* Δ cells the autophagic degradation of GFP-Atg8 was blocked as in *atg1* Δ cells, indicating that Csg2 is essential for autophagy (Fig. 1d).

8) *“The authors overinterpret their results in the paragraph starting with line 190. Here the authors refer to the structural domains of *csg2*, which were investigated with respect to their necessity to control autophagy. The authors state that all ten TM domains are required for the function. This might be so but is not shown by the experiments since no individual – or better - pairwise deletion of the TM regions were performed. The overinterpretation should be avoided but the necessity of the TM domains (the biological significance) for functioning of the channel should be briefly discussed.”*

Response: We thank the reviewer for pointing this. As suggested, we analyzed the necessity of all ten TM domains of Csg2 in autophagy. The TM domain deleted Csg2 mutants (Δ 1-25 for Δ TM1, Δ 390-410 for Δ TM10, Δ TM2 to Δ TM9) were expressed in *csg2* Δ cells and autophagic degradation of GFP-Atg8 was monitored. Results showed that all the TM deleted Csg2 mutants cannot restore GFP-Atg8 degradation in *csg2* Δ cells, suggesting that all the TM domains are necessary for Csg2 function in autophagy (Fig. S2 g-i). We speculated that all the ten TM domains are necessary for Csg2 function as an integrated calcium channel on ER membranes to control calcium homeostasis in ER, which is essential for autophagy activities.

9) “The EF-hand structure is described to contain 40 residues (line 194). What is meant by „residues“? The amino acid range is given from 95 – 107 (Fig S2F), which is 12 amino acids. This should be clarified. It would further be of interest to know whether the Ca-binding EF hand is located in the ER lumen or cytosolically. It should be considered (and discussed) that the Ca-channel Csg2 is regulated by calcium ions.”

Response: We thank the reviewer for pointing this. The “residues” was used to express “amino acids”. To be clear, the “residues” is now changed to “amino acids”. The full EF-hand domain contains amino acids from 85–125. Such information is added in Fig. S2n (previous Fig. S2F). The amino acid fragment from 95–107 (enriched with negatively charged amino acids) is the core region of the EF-hand domain.

To clarify the cytosolic direction of the EF-hand domain, we tagged the EF-hand with GFP by inserting GFP ahead of the EF-hand domain (between amino acid 85 and 86), which is named ⁸⁵-GFP-⁸⁶-Csg2 (Fig. S2j). ⁸⁵-GFP-⁸⁶-Csg2 showed ER distribution as WT Csg2 (Fig. S2k) and restored starvation resistance in *csg2Δ* cells (Fig. S2l). Proteinase digestion protection assays were conducted to analyze the direction of the EF-hand domain. If the EF-hand domain is inside of ER lumen, the fused GFP moiety will not be digested by proteinase K as it is protected by ER membranes, and will be digested at the presence of Triton X-100 that permeabilizes membranes. If the EF-hand domain is located cytosolically, the fused GFP will be digested by proteinase K with or without Triton X-100. Results showed that ⁸⁵-GFP-⁸⁶-Csg2 is digested by proteinase K with or without Triton X-100, the same as cytosolic control protein Ppk1 (Fig. S2m), indicating that the EF-hand is located cytosolically.

To analyze whether Ca²⁺-channel Csg2 is regulated by calcium, *in vitro* channel conductance assays were performed in the presence of different concentrations of calcium. High-frequency ionic current noise (≥ 1 kHz) is correlated with dielectric noise and capacitive noise, mainly caused by low or loss of conductance of channel protein incorporated into lipid membrane. On the other hand, low-frequency fluctuation (≤ 1 kHz) is correlated to the conformational change and high conductance capability of the channel protein. The spectral density (amplitude (pA/Hz)-frequency (Hz)) curve showed that

the current trace of amplitude in the presence of low levels of calcium was different from the control solution in the low-frequency region, but not utterly significant; however, a high level of calcium caused the low frequency region (\leq 1kHz) to be greatly different from the control solution (Fig. S2e). These results indicated that the Ca^{2+} -channel Csg2 is regulated by high levels of calcium.

10) *“l. 146: The authors state that csg2 regulates autophagosome fusion. To my mind a channel does not regulate autophagosome fusion. It rather “is involved in”.”*

Response: There might be a misunderstanding here. We tried to express that Csg2 regulates autophagosome formation (not “fusion”). As suggested, to be clear and accurate, we changed the sentence to “..... suggesting that Csg2 is involved in regulating the formation of autophagosomes.” (line 175 – line 177).

11) *“l. 207/208: The conclusion drawn here do not really reflect the data and results shown in this paragraph, since the question was whether TM and or EF hand motifs are required for csg function.”*

Response: We thank the reviewer for pointing this. The conclusion has been changed to “These data confirm that the TM domains but not the EF-hand domain of Csg2 are required for its function in autophagy regulation.” (line 261 – line 263).

12) *“l. 247 -252: starting with “We speculate that as the calcium probe jGCaMP7c”. First the result should be described and then it should be interpreted. The reasoning given here is not easy to understand. Isn’t it more likely that the overall content of calcium in the cells is comparable but that the distribution within subcellular compartments is different?”*

Response: We thank the reviewer for pointing this. As suggested, we first described and then interpreted the result: “The WT, Csg2 deletion and Csg2 Pmr1 double deletion cells expressing the ER calcium probe jGCaMP7c were investigated and the fluorescence intensity was quantified. The results showed that the increased fluorescence intensity caused by Csg2 deletion was reduced

upon further deletion of Pmr1 (Fig. 4b, c), indicating that the increase in ER calcium caused by Csg2 deletion can be abolished by further Pmr1 deletion.” (line 304 – line 309).

The reasoning was also reinterpreted as suggested: “We speculated that the overall content of calcium in the yeast cells is comparable but that the distribution within subcellular compartments (ER, for example) is different. This is supported by the experimental results. The total amounts of calcium in WT, Csg2 deletion and Csg2 Pmr1 double deletion cells were measured, and the results showed no differences (Fig. S4b).” (line 310 – line 314).

13) *“Could the authors please comment on the experimental data in Fig S3E? In csg2Δ cells, treated with EDTA, the amount of free calcium is primarily reduced in the cytosol, not in the ER. How does it look like after 16h starvation in SD-N medium (which is the experimental setting for the subsequent studies)? It would be also of interest to compare the calcium levels (in Fig S3E) with the calcium levels in the different subcellular compartments in WT yeast cells.”*

Response: We thank the reviewer for these suggestions. First, for Fig. S3E (now Fig. S3g), in the *csg2Δ* cells treated with EDTA, the amount of calcium in ER was reduced (quantified in Fig. S3h), and the cytosol free calcium also seemed to be reduced based on observation of fluorescence intensities. We speculated it may be because that chelator EDTA reduced the amounts of free calcium both in cytosol and in ER.

Second, we added results showing the amounts of calcium in Csg2 deleted cells subject to 16h starvation in SD-N medium, which is similar with that after 3h starvation (Fig. S3g, h).

Third, we compared the calcium levels in vacuoles (Fig. S3k) besides that in ER (Fig. S3g, h). The WT and *csg2Δ* cells were measured on the free calcium levels in vacuoles using a constructed vacuole calcium probe (Cps1-jGCaMP7c), and results showed that the levels of free calcium in vacuole was not affected by Csg2 deletion (Fig. S3k), suggesting that Csg2 mainly affects ER calcium, which is consistent with results showed that Csg2 is a calcium channel localized on ER membranes.

14) *“There is a high amount of information, especially in Fig. 6 and Fig. S6. I would appreciate a more careful description of the experiments, the results and the interpretation – possibly with an additional illustration, in which the results are combined within one scheme. This would largely facilitate the reading of the manuscript and the deciphering of the meaning.”*

Response: We thank the reviewer for this suggestion. Indeed, there is high amount of information in Fig. 6 and Fig. S6 used to indicate that the accumulation of sphingolipid PHS was the reason for autophagy block caused by Csg2 deletion. As suggested, we added an additional illustration (Fig. S6w), trying to combine these results in one scheme. In *csg2Δ* cells, autophagy was blocked; in *csg2Δ tsc10^{ts}* cells, autophagy was restored, indicating that accumulation but not decrease of certain sphingolipid(s) is the reason for autophagy blockage caused by Csg2 deletion. In *csg2Δ lip1^{ts}* or *csg2Δ aur1^{ts}* cells, autophagy was not restored, indicating that it is the accumulation of PHS or DHS (or phos-DHS/PHS) but not the downstream sphingolipid(s) that causes autophagy blockage. Further deletion of Sur2 but not Lcb4/5, Ssc7 or Ccc2 can restore autophagy in *csg2Δ* cells, indicating that it is the accumulation of PHS that causes autophagy blockage in *csg2Δ* cells. The behind reason (disruption of sphingolipid synthase Aur1) for that was then explored in Fig. 7.

15) *“The metabolic link between PHs and Aur1 activity is not clearly presented.”*

Response: We thank the reviewer for pointing this. We measured the PHS levels in WT, *csg2Δ*, and *csg2Δ aur1^{ts}* cells, and results showed that the PHS levels were further greatly increased upon Aur1 disruption (Fig. S6g), indicating the metabolic link between PHS and Aur1.

16) *“l. 410/423: According to the data it should be discussed that Aur1 stability is controlled by prevailing calcium concentrations.”*

Another thesis than the presented ones is the possibly that Aur1 protein forms aggregates in the presence of high calcium levels (aggregate are shown in Fig. 7C) and is subsequently degraded leading to the determined low protein levels of Aur1.

Having analysed the data shown in Fig.3 and Fig.S3 I have the impression that it would be of interest to determine the sensibility of sur2 activity to various calcium concentrations. Possibly, high calcium concentrations activate sur2 resulting in high amounts of PHS and inhibition of autophagy. Vice versa, low amounts inhibit sur2 activity – reducing PHS amounts, which enables autophagy. The addition of PHS again blocks autophagy. Could the authors comment on this hypothesis?”

Response: We thank the reviewer for these suggestions. First, it was suggested that the protein stability of Aur1 is controlled by prevailing calcium concentrations, as calcium chelator restored the protein levels of Aur1 in *csg2Δ* cells (Fig. 7f). This was further supported by the observation that deletion of Pmr1, the P-type ATPase that transports calcium into ER, can restore the protein levels of Aur1 in *csg2Δ* cells after starvation (Fig. S7h).

Second, it is indeed possible that Aur1 protein forms aggregates in the presence of high levels of calcium and is subsequently degraded in *csg2Δ* cells. The puncta distribution of Aur1 in *csg2Δ* cells suggested the aggregate formation of Aur1. We then analyzed the degradation (proteasome or lysosome/vacuole) of Aur1 in *csg2Δ* cells. Proteasome inhibitor MG132 but not vacuole inhibitor NH₄Cl recovered the protein levels of Aur1 in *csg2Δ* cells (Fig. S7p, q), indicating the proteasomal degradation of Aur1.

Third, the hypothesis (Possibly, high calcium concentrations activate sur2 resulting in high amounts of PHS and inhibition of autophagy. Vice versa, low amounts inhibit sur2 activity – reducing PHS amounts, which enables autophagy.) proposed by the reviewer is interesting. We then analyzed the cellular localization and protein levels of Sur2 at the presence of high ER calcium (by Csg2 deletion) or low ER calcium (by Pmr1 deletion). Results showed that Csg2 deletion had no effect on the cellular localization and protein levels of Sur2, while Pmr1 deletion caused upshift of Sur2 protein bands in WB gels (Fig. S7k, l). We speculated that low ER calcium (by Pmr1 deletion) disturbs Sur2, reducing PHS amounts and promoting autophagy (as proposed by the reviewer), which is in line with the results that Pmr1 deletion promotes autophagy (Fig. S1k). High ER calcium (by Csg2 deletion), however, showed no effect on the cellular localization and protein levels of Sur2, indicating that

high ER calcium may disrupt Aur1 but not Sur2 to cause increase of PHS and blockage of autophagy. Since many sphingolipid enzymes are localized at ER, maybe they are subject to regulation by calcium although differently. For instance, Sur2 may be disturbed by low calcium but tolerate high calcium, while Aur1 is disrupted by high calcium. In the future, it will be interesting to systematically analyze the activities of sphingolipid enzymes regulated by calcium and/or by other ions.

17) *“In the Discussion l. 446-448: To my mind here the authors contradict the data shown in Fig 5C’.”*

Response: We thank the reviewer for pointing this. Here (previous l.446-448) we proposed potential mechanism behind PHS inhibiting autophagy. To be clear and accurate, we now delete this speculation description.

18) *“In Fig. 3. The authors state in the legend D) “transportation of RFP-Ape1 was observed In atg1Δ cells“. It was investigated but not observed !”*

Response: We thank the reviewer for pointing this. The word “observed” is changed to “investigated”.

19) *“Due to language problems some of the paragraphs/sentences are difficult to understand, sometimes the meaning remains obscure.*

Legend to Fig. 4A: should be rephrased

In general, there are language problems and at some passages it is almost not possible to catch the meaning. Editing by a native speaker is required.

Just to list some examples:

Sentence l. 50 -54: „... by enabling calcium influx into and out of the environment,...“

l. 128: should be rephrased, for example: “The degradation of GFP-50Q, another..., was shown to be blocked in csg-depleted yeast cells”.

l. 199: "... showed no adverse effect on ...”

rephrasing also required l. 214, l. 224, l. 239 – 240, l. 242, l. 247-252, l.302/303, l. 345-350, l. 394-397.

l. 408: the term “necessary” should be “physiological” or “adequate”.”

Response: We thank the reviewer for pointing these language problems. As suggested, the manuscript is edited by a native English speaker from the professional editing service American Journal Experts (<https://www.aje.com/>, Order No. PK2NNN25). The Legend to Fig. 4A and sentences I.50-54, I.128, I.199, I.214, I.224, I.239–240, I.242, I.247-252, I.302/303, I.345-350 and I.394-397 are rephrased or especially edited (I.53-57, I.159-162, I.253-255, I.268-270, I.279-281, I.300-302, I.302-304, I.310-314, I.371-372, I.425-427 and I.519-521 in the revised manuscript). The term “necessary” in I.408 is changed with “physiological” (I.532 in the revised manuscript).

Reviewer #2

We thank the reviewer for the time and the positive comments “*This is a very thorough investigation into the topic of calcium accumulation and autophagy. One major strength of this manuscript is that the authors are careful to use multiple assays to validate individual observations. This increases the overall confidence in the data presented.*”.

The specific concerns are addressed as shown below.

Major concerns:

1) “1) *The authors demonstrate that the deletion of $csg2\Delta$ leads to a block of autophagy in response to nitrogen starvation. Do the authors believe that this block of autophagy only occurs in response to starvation (nitrogen or glucose)? There are several forms of basal autophagy that go on during times when cells have access to nutrients including the Cvt form of selective autophagy. Therefore, it would be informative for the authors to investigate Ape1 processing when nutrients are plentiful to determine if autophagy is blocked even prior to nutrient starvation.*”

Response: We thank the reviewer for this suggestion. Blockage of autophagy in $csg2\Delta$ cells has been shown in the situations of nitrogen starvation (Fig. 1d) and glucose starvation (Fig. S1a). We then analyzed the Cvt form of autophagy in nutrient-rich conditions by investigating Ape1 processing. Results showed that Ape1 processing in $csg2\Delta$ cells is the same as that in WT cells in nutrient-rich conditions (Fig. S1j), indicating that autophagy is not blocked by Csg2 deletion prior to nutrient starvation.

2) “2) *The authors do not adequately explain the Electrophysiology Measurement Assay or the Cell ER Calcium Concentration Fluorescence Detection Kit assay in enough detail. The authors should include more details for these assays in the methods. In particular, the authors should provide more details as to what is meant by “after the protein was embedded 1ml of solution was changed ...” on line 565 and 566. Is the Cell ER Calcium Concentration Fluorescence Detection Kit assay a microscopy-based assay? If so, the authors should include representative microscopy images.*”

Response: More detailed information is added to describe methods “Electrophysiology Measurement Assay” and “Cell ER Calcium Concentration Fluorescence Detection Kit assay” in the Methods section (line 730 - line 765 and line 793- line 827).

The sentence “*after the protein was embedded 1ml of solution was changed ...*” is now corrected to “After the protein was embedded, the solution was changed to 1 mL of solution without proteins.” in the new description of this assay (line 739 - line 740). The Cell ER Calcium Concentration Fluorescence Detection Kit assay is a fluorescence spectrophotometer-based assay (excitation wavelength at 490 nm and emission wavelength at 525 nm).

3) *“3) Based on the model proposed by the authors one would predict that adding calcium to yeast cells should lead to an inhibition of autophagy in WT cells in SD-N. However, in Figure 3A this is not what the authors observe, and the authors do not adequately acknowledge this. Do the authors have any speculation as to why they do not observed autophagy inhibition when calcium is added to WT cells? In addition, given the results from this figure the authors may want to consider changing the title of the figure as adding calcium did not lead to a blockage of autophagy in WT cells. It may also be helpful to monitor calcium accumulation by stt3-jGCaMp7c in WT cells in the absence and presence of added calcium as this may show if there is an increase in ER calcium when CaCl₂ is added and may help explain their results.”*

Response: We speculated that WT cells can resist added calcium by maintaining the intracellular calcium homeostasis, which means that the ER calcium in WT cells is not affected when adding calcium. Various calcium channels may contribute to the calcium homeostasis.

As suggested, the title of the figure is changed to “Figure 3. The blockage of autophagy in Csg2 deleted cells was related with the accumulation of ER calcium.”. ER calcium in WT cells was monitored by stt3-jGCaMp7c in the absence and presence of added calcium. Results showed that the ER calcium in WT cells was not affected by added calcium (Fig. S3b). This is in line with the results that added calcium cannot inhibit autophagy in WT cells (Fig. S3a).

4) “4) In a similar comment to point 3. When the authors add PHS to WT cells in figure 6 it does not lead to a blockage of autophagy. It seems that given the model this would be predicted to be the case. Do the authors have any speculation on why this might be the case? The authors should acknowledge this result in the manuscript and clearly describe in their model that they do not observe autophagy inhibition in WT cells when calcium or PHS is added.”

Response: We thank the reviewer for this comment. As suggested, we described in the mechanism model that autophagy is not inhibited in WT cells when calcium or PHS is added. We speculated that the added PHS into WT cells may not cause significant change on intracellular PHS levels, therefore, the autophagy in WT cells was not inhibited when PHS was added. We then experimentally investigated the PHS levels in WT cell when PHS was added, and results showed that the intracellular PHS levels were not affected by added PHS (Fig. S6n). We speculated that the possible reason is that the sphingolipid metabolic enzymes in WT cells function to promote the efficient metabolism of added PHS, leading to steady levels of intracellular PHS. This was supported by the results showed that added PHS caused significant increase of intracellular PHS in *csg2Δ sur2Δ* cells (Fig. S6o), in which situation the PHS upstream enzyme Sur2 is deleted and the downstream enzyme Aur1 is disrupted by accumulated ER calcium caused by Csg2 deletion.

5) “5) In Figure 6F the authors knockout *Lcb4* and *Lcb5* to test the role of different enzymes on autophagy inhibition. Would the authors expect that a deletion of *Lcb4* and *Lcb5* should lead to an accumulation of PHS? Since *Lcb4* and *Lcb5* use PHS as a substrate one would predict that this might be the case. If *Lcb4* and *Lcb5* knockout does lead to an increase in PHS, why does deletion of *Lcb4* and *Lcb5* in WT cells not lead to a block of autophagy?”

Response: We thank the reviewer for this comment. To figure out why the deletion of *Lcb4* and *Lcb5* showed no autophagy inhibition, we measured PHS levels in *lcb4Δ lcb5Δ* cells. Results showed that PHS showed no significant change in *lcb4Δ lcb5Δ* cells compared with WT cells (Fig. S6e), which potentially explained why autophagy was not inhibited by deletion of *Lcb4* and

Lcb5. We speculated that either the step of synthesis of PHS-1P from PHS is at low rate or the increased PHS caused by deletion of Lcb4 and Lcb5 is efficiently utilized for synthesis of PHCer, leading to no significant changes on PHS levels in *lcb4Δ lcb5Δ* cells.

Reviewer #3

We thank the reviewer for the time and positive comments “*The experiments have been carried out well and were reported clearly. There is information in the work that will be of interest to the field of autophagy. The manuscript presents strong evidence for an important role of ER calcium channels in regulation of autophagy.*”.

Specific comments are addressed as shown below.

Major comments:

1) “1. As mentioned above, it remains possible that PHS affects fusion with the vacuole or vacuolar acidification. It should be determined which steps are inhibited by PHS.”

Response: We thank the reviewer for this suggestion. First, we used autophagosome marker GFP-Atg8 to analyze the steps of autophagy inhibited by PHS. Inhibition of autophagosome formation will cause diffuse distribution of GFP-Atg8; inhibition of autophagosome fusion will cause accumulation of dots of GFP-Atg8; inhibition of autophagosome degradation will cause accumulation of GFP-Atg8 inside of vacuoles. Addition of PHS inhibited autophagy in *csg2Δ*+EDTA cells (Fig. 6l), and in such condition GFP-Atg8 showed diffuse cytosolic distribution (Fig. S6r), which indicates that PHS inhibited the upstream step of autophagy, autophagosome formation, but not the downstream fusion or vacuole degradation steps.

Second, we investigated the vacuole transportation of endocytosis marker GFP-Sna3 at the presence of added PHS, and results showed that its trafficking into vacuole maintained normal (Fig. S6s), indicating the vesicle-vacuole fusion was not affected by PHS.

Third, we analyzed the vacuole degradation of GFP-Sna3 (indicated by appearance of GFP moiety) when PHS added, and results showed that the vacuole degradation of GFP-Sna3 was also maintained normal (Fig. S6t), suggesting that vacuolar acidification was not affected by PHS.

2) “2. Based on the results with *tsc10*, *lip1*, and *aur1* mutant strains and double mutants with *csg2* Δ , the authors claim that accumulated PHS causes blockage of autophagy after starvation. However, the manuscript does not provide any experimental evidence to confirm the levels of PHS and other sphingolipids. Lipid analysis needs to be done as this is directly relevant to the conclusion.”

Response: We thank the reviewer for this suggestion. We analyzed the levels of PHS in yeast cells with *tsc10^{ts}*, *lip1^{ts}*, and *aur1^{ts}* double mutation with *csg2* Δ . Results showed that *tsc10^{ts}* double mutation with *csg2* Δ caused decrease of PHS compared with that in *csg2* Δ cells, while *lip1^{ts}* or *aur1^{ts}* double mutation with *csg2* Δ caused increase of PHS (Fig. S6g). This was in line with the positions of Tsc10, Lip1 and Aur1 in the process of sphingolipid synthesis (Fig. 6c), with Tsc10 at the upstream of PHS for its synthesis and Lip1 and Aur1 at the downstream of PHS for its metabolism. The low levels of PHS in *tsc10^{ts}* *csg2* Δ cells and high levels of PHS in *lip1^{ts}* *csg2* Δ or *aur1^{ts}* *csg2* Δ cells were in line with the results that autophagy in *tsc10^{ts}* *csg2* Δ cells was restored but autophagy in *lip1^{ts}* *csg2* Δ or *aur1^{ts}* *csg2* Δ cells was blocked (Fig. 6e).

3) “3. Since *Csg2* has been shown to play an essential role in MIPC synthesis, the manuscript should provide strong experimental evidence to eliminate the possibility that reduced MIPC levels in *csd2* Δ cells cause blockage of autophagy. Further analysis needs to be done. For example, *rsb1* deletion mutant (Kihara and Igarashi Y, 2002) or *ypc1* mutant (Mao et al., 2000) strain combined with *lcb4* and *lcb5* mutations that does not affect MIPC synthesis but is expected to increase PHS levels might be used for the analysis. In addition, if the authors can show that the autophagy defect are restored in *csg2* Δ background with normal levels of IPC but reduced levels of PHS, then the possibility that increased IPC levels in *csg2* Δ cells cause blockage of autophagy would be also ruled out, although it may be difficult to create such lipid level conditions.

Csg1 and *Csh1* function in MIPC synthesis as catalytic subunits (Uemura et al., 2003). Although *csg1* Δ *csh1* Δ double mutant exhibits a calcium sensitivity, this sensitivity has been proposed to be caused by an accumulation of IPC-C, one of IPC classes. If ER calcium accumulation in *csg2* Δ cells directly causes

reduced level of Aur1 protein and increases the level of PHS, csg1 Δcsh1Δ double mutant may not affect Aur1 stability. If the authors can show a specific role of Csg2 and that Csg1 and Csh1 are not involved in autophagy regulation, the data will be more convincing evidence to support the proposal.”

Response: We thank the reviewer for these suggestions. First, we analyzed whether the reduction of MIPC can cause blockage of autophagy. Mutation of Tsc10, the very upstream enzyme of sphingolipid synthesis process, showed no effect on autophagy (Fig. 6b). Further, tsc10^{ts} double mutation with csg2Δ restored autophagy (Fig. 6e). These results indicated that reduction of sphingolipids (including MIPC) was not the reason for autophagy blockage in csg2Δ cells. This was in line with the results that addition of PHS, which would increase the levels of sphingolipids, caused blockage of autophagy (Fig. 6j-l). Therefore, we speculated that the possibility of reduction of MIPC causing blockage of autophagy in csg2Δ cells can be ruled out.

Second, we analyzed the effects of Rsb1 or Ypc1 deletion on autophagy in lcb4Δ lcb5Δ cells. Results showed that deletion of Rsb1 or Ypc1 showed no effect on autophagy in lcb4Δ lcb5Δ cells (Fig. S6m). We speculated that either the step of synthesis of PHS-1P from PHS (catalyzed by Lcb4 Lcb5) is at low rate or the increased PHS caused by deletion of Lcb4 and Lcb5 is efficiently utilized for synthesis of PHCer, leading to no significant changes on PHS levels in lcb4Δ lcb5Δ cells (Fig. S6e). Therefore, further deletion of Rsb1 or Ypc1 in lcb4Δ lcb5Δ cells showed no effect on autophagy (Fig. S6m).

Third, we analyzed the possibility that the increased IPC causes blockage of autophagy in csg2Δ cells. If this is the case, then Aur1 (IPC synthase) mutation in csg2Δ cells should restore autophagy, as Aur1 mutation would reduce the levels of IPC. However, results showed that Aur1 mutation in csg2Δ cells did not restore autophagy (Fig. 6e). Therefore, we speculated that the possibility of increased IPC causing blockage of autophagy in csg2Δ cells can be ruled out.

Fourth, we analyzed the roles of Csg1 (also named Sur1) and Csh1 in autophagy. Results showed that double deletion of Csg1 and Csh1 caused blockage of autophagy (Fig. S6d) and increase of PHS levels (Fig. S6e). The blocked autophagy in csg1Δ csh1Δ cells can be restored by further deletion of

Sur2 (whose deletion would reduce PHS) (Fig. S6l). Furthermore, double deletion of Csg1 and Csh1 showed no effect on the protein levels or the cellular distribution of Aur1 (Fig. S7i, j). We speculated that although both Csg2 deletion and Csg1/Csh1 deletion caused blockage of autophagy, the behind mechanisms are different. Deletion of Csg2, the ER calcium channel, induced the accumulation of calcium in ER, leading to disruption of Aur1 and followed increase of PHS that caused blockage of autophagy. Deletion of Csg1/Csh1, the enzymes in sphingolipid synthesis pathway, induced the increase of PHS and further blockage of autophagy. These results supported the hypothesis that increased PHS causes blockage of autophagy.

4) *“4. Does the decrease in Aur1 protein level in the csg2 Δ strain occur only during starvation? What about in nutrient rich media? Is it degraded by the ubiquitin (Ub)-proteasome pathway or in vacuoles? Do the protein levels of Aur1 in the csg2 pmr1 double mutant strain remain normal?”*

Response: We thank the reviewer for these questions. First, the protein levels of Aur1 were analyzed under nutrient rich conditions. Results showed that the protein levels of Aur1 in *csg2 Δ* cells were similar as that in WT cells in nutrient rich conditions and were reduced after starvation (Fig. S7h).

Second, we analyzed the degradation (proteasome or vacuole) of Aur1 in *csg2 Δ* cells in starvation conditions. Results showed that proteasome inhibitor MG132 but not vacuole inhibitor NH₄Cl recovered the protein levels of Aur1 in *csg2 Δ* cells (Fig. S7p, q), indicating the proteasomal degradation of Aur1 in *csg2 Δ* cells in starvation conditions.

Third, we analyzed whether deletion of Pmr1, the high-affinity P-type ATPase mediating calcium transport into the ER, can restore the protein levels of Aur1. We detected the protein levels of Aur1 in Csg2 Pmr1 double deletion cells. Results showed that Aur1 maintained similar protein levels in *csg2 Δ pmr1 Δ* cells as that in WT cells (Fig. S7h), indicating that it was the accumulation of calcium in ER (caused by Csg2 deletion) that caused disruption of Aur1 proteins.

5) *“5. Aur1 and Kei1 has been well known to localize to the Golgi apparatus (Levine et al., 2000; Lisman et al., 2004; Sato et al., 2009). Is the ER localization of Aur1 and Kei1 shown in this study a starvation-specific phenomenon? Aur1 and Kei1 localization in nutrient rich media should be shown. It is also necessary to analyze whether the dot-like compartments of Aur1-GFP in the csg2Δ cells are Golgi apparatus.”*

Response: We thank the reviewer for these questions/suggestions. First, we analyzed the localization of Aur1 and Kei1 in nutrient rich media. Results showed that in nutrient rich conditions (SD-N 0h), both Aur1 and Kei1 exhibited merge distribution with ER marker, which was not affected by Csg2 deletion (Fig. S7d, e). We speculated that Aur1 and Kei1 may localize both at ER and/or Golgi, and in different yeast background or culture situations, their localizations vary. This may explain the subcellular location of Aur1 is labeled Golgi localization in UniProt Annotation while labeled ER and Golgi localization in GO Annotation (https://www.uniprot.org/uniprotkb/P36107/entry#subcellular_location).

Second, we analyzed whether the dot-like compartments of Aur1-GFP in the csg2Δ cells are Golgi apparatus. Aur1-GFP was expressed in csg2Δ cells together with Golgi marker (Sec7) and other organelle markers (mitochondria marker Cox4 and lipid droplet marker Tgl3) used as controls. Localization of Aur1-GFP was investigated after starvation. Results showed that Aur1-GFP dots were colocalized with Golgi marker, but not mitochondria or lipid droplet markers (Fig. S7o), indicating that the dot-like compartments of Aur1-GFP in the csg2Δ cells are Golgi apparatus.

6) *“6. If Aur1 relocates from the Golgi to the ER during starvation, sphingolipid metabolism will be severely affected by starvation. Figure 5 shows the difference in lipid levels between WT and csg2Δ strains after starvation and the data before starvation are missing. The differences in lipid levels before and after starvation, as well as between WT and csg2Δ strains before starvation should be analyzed to determine if the increase in PHS and ceramide levels in csg2Δ cells is specific to starvation.*

A sentence in ABSTRACT “Deletion of Csg2 causes increases in calcium levels in the ER and then disturbs Aur1 stability, leading to accumulation of the

bioactive sphingolipid phytosphingosine.....” will mislead the reader. Therefore, it needs to be clarified whether changes in lipid levels, as well as Aur1 stability and localization are specific to starvation or not.”

Response: We thank the reviewer for these suggestions. First, we analyzed the PHS levels before starvation. Results showed that the levels of PHS in *csg2Δ* cells were similar as that in WT cells when yeast cells were cultured in nutrient rich conditions (Fig. S5h), indicating that PHS increase in *csg2Δ* cells was specific to starvation. This was in line with the results that disruption of Aur1 induced by Csg2 deletion was also specific to starvation (Fig. S7h).

Second, the Aur1 stability and localization in *csg2Δ* cells were analyzed before starvation. Results showed that the changes were not found when cells were cultured in nutrient rich conditions (Fig. S7d, h). We changed the sentence in ABSTRACT to “Under starvation conditions, deletion of Csg2 causes increases in calcium levels in the ER and then disturbs Aur1 stability, leading to accumulation of the bioactive sphingolipid phytosphingosine.....”.

7) *“7. The authors claim that accumulated PHS causes blockage of autophagy after starvation. Explain why the addition of PHS to WT cells does not inhibit autophagy (Figure 6J and L).”*

Response: We thank the reviewer for this comment. Indeed, we noticed that autophagy was not inhibited in WT cells when PHS was added. We speculated that the added PHS into WT cells may not cause significant change on intracellular PHS levels, therefore, the autophagy in WT cell was not inhibited. We then experimentally investigated the PHS levels in WT cell when PHS was added and results showed that the intracellular PHS levels were not affected by added PHS (Fig. S6n). We speculated that the sphingolipid metabolic enzymes in WT cells function to promote the efficient metabolism of added PHS, leading to steady levels of intracellular PHS. This was supported by the results showed that added PHS caused significant increase of intracellular PHS in *csg2Δ sur2Δ* cells (Fig. S6o), in which situation the PHS upstream enzyme Sur2 was deleted and the downstream Aur1 was disrupted by accumulated ER calcium caused by Csg2 deletion.

8) “8. Exogenously added PHS is linked to de novo synthesis pathway via phosphorylation by Lcb4/Lcb5 and dephosphorylation by Lcb3 processes (Funato et al., 2003). Therefore, exogenously added PHS is expected to not block autophagy in *csg2Δsur2Δlcb3Δ* or *csg2Δpmr1Δlcb3Δ* (or not in *lcb3Δ* single mutant if the addition of PHS to WT cells inhibits autophagy). To confirm the importance of PHS in autophagy regulation, the effects of *lcb3* deletion should be shown.”

Response: We thank the reviewer for this suggestion. We analyzed autophagy in *lcb3Δ*, *csg2Δsur2Δlcb3Δ* or *csg2Δpmr1Δlcb3Δ* cells when exogenous PHS was added. Autophagy in *lcb3Δ* cells was normal when PHS was added (Fig. S6u). We speculated that disturbing the step of phosphorylation of PHS into PHS-1P (catalyzed by Lcb4/Lcb5 and reversed by Lcb3) caused little effect on PHS levels, which thus showed no effect on autophagy. This was supported by the results that deletion of Lcb4 and Lcb5 showed no autophagy inhibition (Fig. 6f) and PHS levels in *lcb4Δ lcb5Δ* cells were not changed (Fig. S6e). Deletion of Sur2 or Pmr1 restored autophagy in Csg2 deleted cells, and this restoration was abolished when PHS was added (Fig. 6j, k). Deletion of Lcb3 in *csg2Δsur2Δ* or *csg2Δpmr1Δ* cells cannot restore autophagy when PHS was added (Fig. S6v), suggesting that Lcb3 showed no effect on autophagy.

9) “9. It was not tested whether the function of Aur1 is disrupted by increase in PHS. Does PHS addition affect Aur1 protein level and localization in *csg2Δsur2Δ* or *csg2Δpmr1Δ*, as well as in WT cells?”

Response: We thank the reviewer for this suggestion. We analyzed Aur1 protein levels and localization in WT, *csg2Δsur2Δ* or *csg2Δpmr1Δ* cells. Results showed that the protein levels and localization of Aur1 were normal in WT cells and in *csg2Δpmr1Δ* cells (ER calcium not increased), and added PHS showed no effect on that (Fig. S7s, t). The protein levels and localization of Aur1 were disrupted in *csg2Δsur2Δ* cells (ER calcium increased) and added PHS showed no further effect on that (Fig. S7s, t). Therefore, we speculated that the protein levels and localization of Aur1 were regulated by ER calcium but not PHS.

10) “10. The authors argue that calcium accumulation causes blockage of autophagy through dysfunction of Aur1. If so, Aur1 overexpression or artificial localization (for example, utilizing GFP-GFP-binding protein (GBP)) may suppress blockage of autophagy. It can be tested.”

Response: We thank the reviewer for this suggestion. We overexpressed Aur1 using high-copy vector (2 μ M vector, p425-prADH1-Aur1-HA) in *csg2 Δ* cells and results showed that blockage of autophagy was not suppressed (Fig. S7r). The reason we speculated was that the actual protein levels of Aur1 was kept at low levels (Fig. S7r) because of ER calcium accumulation in *csg2 Δ* cells.

Minor comments:

1) “1. In Fig S3E, is dot-like compartments of ER calcium probe Stt3-jGCaMP7c in *csg2 Δ* with CaCl₂ the ER? DIC images of cells with CaCl₂ are different from others. Is cell death beginning to occur?”

Response: First, we analyzed whether the dot-like compartments of ER calcium probe Stt3-jGCaMP7c in *csg2 Δ* cells with CaCl₂ were ER. Result showed that the dots of Stt3-jGCaMP7c were colocalized or close with ER marker (Fig. S3i). We speculated that under situations of very high levels of calcium (by deletion of calcium channel Csg2 and further exogenously added calcium), ER was partially compartmentalized for store of calcium.

Second, we analyzed whether these cells in such conditions were dead cells or beginning to die. In Fig. S3E (now Fig. S3g in revised manuscript), cells were treated with SD-N and added CaCl₂ for 3 hours. Same time (3h) and longer time (16h, 24h) points of treatments were taken and cell viabilities were analyzed by spotting the cells after treatment to YPD plates and their growth was detected. Results showed that the cell viabilities were not affected (Fig. S3j), indicating that the *csg2 Δ* cells treated with SD-N and CaCl₂ (3h) were not dead or beginning to die.

2) “2. In Fig S6B, the enzyme that converts PHCer to PHS is Ypc1, not Ydc1.”

Response: We thank the reviewer for pointing this mistake. The “Ydc1” enzyme that converts PHCer to PHS was corrected to “Ypc1” in Fig. S6b.

3) “3. In the third paragraph of the Discussion, the author states that “although we established the link between PHS and autophagy inhibition at the autophagosome formation stage, how this inhibition is achieved is unclear.”. However, as noted in major comments 1, the authors have not examined which steps are inhibited by PHS. It should be expressed accurately.”

Response: We thank the reviewer for this suggestion. As mentioned in response to comment 1 (Major comments), we tried to analyze the steps of autophagy inhibited by PHS. First, we used autophagosome marker GFP-Atg8 to analyze the steps of autophagy being inhibited by PHS. Inhibition of autophagosome formation will cause diffuse distribution of GFP-Atg8; inhibition of autophagosome fusion will cause accumulation of dots of GFP-Atg8; inhibition of autophagosome degradation will cause accumulation of GFP-Atg8 inside of vacuoles. Addition of PHS inhibited autophagy in *csg2Δ*+EDTA cells (Fig. 6l), and in such condition GFP-Atg8 showed diffuse cytosolic distribution (Fig. S6r), indicating that PHS inhibited the upstream step of autophagy, autophagosome formation, but not the downstream fusion or vacuole degradation steps. Second, we investigated the vacuole transportation of endocytosis marker GFP-Sna3 at the presence of added PHS, and results showed that its trafficking into vacuole was normal (Fig. S6s), indicating the vesicle-vacuole fusion was not affected by PHS. Third, we analyzed the vacuole degradation of GFP-Sna3 (indicated by appearance of GFP moiety), and results showed it was normal (Fig. S6t), suggesting that vacuolar acidification was not affected by PHS. These results suggested that PHS inhibited the autophagosome formation but not autophagosome fusion or degradation steps of autophagy process.

Reviewer #4

We thank the reviewer for the time and positive comments “*Most experiments have been carried out well and have the appropriate controls. The manuscript is also written well and clearly.*”.

The specific points/comments are addressed as shown below.

Major points:

1) “ - *The lipidomics data have to be completely revised. From the material and methods section it is not clear what was measured, which quality controls were used and how the lipid species shown were selected. To the best of my knowledge yeast has only saturated and mono-unsaturated fatty acids. Why are lipid species such as PC 36:5 shown in the analysis. Why is cholesterol part of the analysis when yeast only produces ergosterol as the major sterol. In which ion mode where the data obtained. Some of the species are mostly detectable in positive ion mode (TAGs, DAGs, PC) other are better detected in negative ion mode (PI, PS, PG). Why where the lipids presented chosen. In the results table there needs to be an annotation of the Q1 and Q3 masses used to determine the shown lipid species. In addition, the retention times of the standards and the chosen lipid species need to be presented. I would recommend to include the checklist from a recently published paper (doi: 10.1038/s42255-022-00628-3) here. Since this was also published by the nature publishing group this should be incorporated here.*”

Response: We thank the reviewer for these suggestions. First, we apologize for the insufficiency in our method description. Using the guidelines provided in the comment paper (doi:10.1038/s42255-022-00628-3) brought forth by the reviewer, which has been cited in our revised manuscript, we have revised our materials and methods pertaining to lipidomics analysis, as follows.

“Lipidomics analysis of yeast cells was conducted at LipidALL Technologies using an MRM library constructed based on the company’s preceding publications on yeast lipidomes¹⁴⁶⁻¹⁴⁸. Lipids were extracted from yeast cells using standard Bligh and Dyer’s method as described previously¹⁴⁹. Briefly, yeast cells were homogenized in 750 μ L of chloroform:methanol 1:2 (v/v) containing 10% deionized water with glass beads on an automated bead shaker

(OMNI, USA). The homogenate was then incubated at 1500 rpm for 1 h at 4°C. At the end of the incubation, 350 µL of deionized water and 250 µL of chloroform were added to induce phase separation. The samples were then centrifuged and the lower organic phase containing lipids was extracted into a clean tube. Lipid extraction was repeated once by adding 500 µL of chloroform to the remaining tissues in aqueous phase, and the lipid extracts were pooled into a single tube and dried in the SpeedVac under OH mode. Samples were stored at -80°C until further analysis.

Lipidomics methodology was reported according to standard guidelines¹¹³. Polar lipids including the classes of PE, PA, PI, PS, PG, LPE, LPA, LPI, LPS and FFAs were analyzed in the negative ion mode under electrospray ionization (ESI) on an Agilent 1260 HPLC coupled to Sciex 5500 QTRAP, with source parameters as follows, CUR 10, TEM 400°C, GS1 20, GS2 20. Other polar lipid classes such as PC, LPC, Sph and PhytoCer were analyzed in the ESI positive ion mode on the same machine, with source parameters CUR 10, TEM 400°C, GS1 30, GS2 30. Separation of individual lipid class of polar lipids was carried out using a Phenomenex Luna 3µm-silica column (internal diameter 150 × 2.0 mm) with the following conditions: mobile phase A (chloroform: methanol: ammonium hydroxide, 89.5:10:0.5) and mobile phase B (chloroform: methanol: ammonium hydroxide: water, 55:39:0.5:5.5). The gradient started with 2% B, which was maintained for 1 min before increasing to 35% B over the next min. %B was further increased to 65% over the next 5 min, and finally reached 100% B at the 8th min. The gradient was maintained at 100% B for 5 min, and then decreased back to 2% B and equilibrated for 4 min prior to the next injection. Flow rate was 350 µL/min and column oven temperature was at 35°C. Individual lipid species were quantified by referencing to spiked internal standards, which included d9-PC32:0(16:0/16:0),d7-PE33:1(15:0/18:1),d31- PS,d7-PG33:1(15:0/18:1),d7-PI33:1(15:0/18:1),d7-PA33:1(15:0/18:1),Cer(d18:1-d7/15:0),d7-LPC18:1,d7-LPE18:1,C17:0-LPA,C17:1-LPI,C17:1-LPS,d17:1-Sph from Avanti Polar Lipids. Inc. Free fatty acids were quantitated using d31-16:0 (Sigma-Aldrich) and d8-20:4 (Cayman Chemicals) as internal standards. Glycerol lipids including diacylglycerols (DAG) and triacylglycerols (TAG) were quantified using a modified version of reverse

phase HPLC/MRM on an Agilent 1260 coupled to SCIEX QTRAP 5500 under ESI positive ion mode¹⁴⁸. Separation of neutral lipids were achieved on a Phenomenex Kinetex-C18 2.6 μm column (i.d. 4.6x100 mm) using an isocratic mobile phase containing chloroform:methanol:0.1 M ammonium acetate 100:100:4 (v/v/v) at a flow rate of 170 μL for 17 min. Relative quantities of TAGs were calculated by referencing to spiked internal standard TAG (16:0)₃-d₅ obtained from CDN isotopes, while DAGs were quantified using d₅-DAG18:1/18:1 from Avanti Polar Lipids. Free cholesterol and cholesteryl esters were analyzed under atmospheric pressure chemical ionization mode in the positive polarity on an Agilent 1260 HPLC coupled to SCIEX QTRAP 5500 as described previously, with d₆-cholesterol and d₆-C18:0-cholesteryl ester (CE) (CDN isotopes) as internal standards¹⁴⁷.

Quality control (QC) samples were prepared from pooled lipid extracts of individual yeast samples, and aliquots of QC samples were injected at the beginning and at the end of the batch run, which comprised the biological samples from each group. Lipids were considered detectable if signal-to-noise ratio was above 3 in QC samples, which were representative of the biological samples. The quality control samples showed good consistency (Fig. S5e).

We agree with the reviewer the C16:1 and C18:1 constitute the major unsaturated fatty acids in *Saccharomyces cerevisiae* under normal growth conditions. Nonetheless, yeast cells can take up exogenous polyunsaturated fatty acids from the culture medium and incorporate these fatty acids into their lipidomes (Grillitsch et al., 2011; Suomalainen and Keränen, 1968), even when these impurities may present in traces. Coupled with the high sensitivity of our MRM approach used in this assay, we speculate that PC 36:5 may arise from fatty acid impurities present in the growth process. We present two published examples below, which demonstrates the presence of polyunsaturated fatty acids in yeast due to fatty acid impurities in culture medium, as follow.

1 (Fig. 2 in Grillitsch *et al*, 2011- PMID: 21820081), note the presence of TAG 54:6, 54:5 in yeast cells, implying that one of the constituent fatty acids must contain more than one C=C bonds in their structures.

TABLE 3

The effect of pure grade oleic acid and its ethyl ester on the fatty acid composition of baker's yeast in a medium lacking biotin but containing aspartic acid.

Addition	Oleic acid		Ethyl ester of oleic acid	
Yeast yield, g d.m.	22.3		16.9	
Fatty acids, % of yeast dry matter in total yeast mass, mg	2.10		1.65	
Fatty acid composition, % and the main components in total yeast mass, mg	468		279	
	%	mg	%	mg
C ₁₀ -C ₁₂	2.9	13.6	5.5	15.3
C ₁₃	0.6	2.8	0.7	2.0
C ₁₄	2.0	9.4	2.3	6.4
C _{14:1}	0.4	1.9	2.9	8.1
C ₁₅	0.4	1.9	0.2	0.6
C _{15:1}	-	-	0.3	0.8
C ₁₆	15.1	70.7	6.0	16.7
C _{16:1}	21.5	101	49.9	139
C _{16:2}	0.5	2.3	-	-
C ₁₇	1.3	6.1	-	-
C _{17:1}	-	-	0.4	1.1
C ₁₈	2.7	12.6	0.8	2.2
C _{18:1}	41.9	196	29.1	81.2
C _{18:2}	8.6	40.2	1.5	4.2
C _{18:3}	2.1	9.8	0.4	1.1

Growth in 11 I for 6 hr at 30°C with 9 g A₂ stage yeast as seed yeast. Oleic acid added, as acid or ester, 5 g. Fatty acid composition of yeast determined by gas chromatography of the methyl esters.

2 (Table 3 in Suomalainen *et al*, 1968- [https://doi.org/10.1016/0009-3084\(68\)90006-6](https://doi.org/10.1016/0009-3084(68)90006-6)), note the presence of very low level of C18:3 fatty acids in the fatty acid profiles of yeast cells, likely attributed to impurities in culture medium.

We present below our XICs for the series of C36-PCs from our data, which clearly showed a detectable peak for PC 36:5, despite being present in relatively very low quantity compared to major species such as PC 36:2 and PC 36:1.

Nonetheless, we recognize this concern from the reviewer that these low abundant species likely resulting from background impurities should not be presented in the final data to avoid confusion to the readers. As such, we have removed this species and re-generated the corresponding diagrams in our revision. Nonetheless, we wish the reviewer to note that our general PC profile was in good agreement with preceding knowledge on lipidome profile in yeast, with PC 32:1, PC32:2, PC 34:1 and PC 34:2 forming the major species (presented below).

As we had expressed our lipidome data in μmol lipids/mg protein, removing these low abundant impurities does not affect our results and the major conclusions of our work.

As for cholesterol, ergosterol is indeed the major sterol in yeast cells in our analysis. The mean ergosterol level was at 0.067 μmol /mg protein, while that of cholesterol was at 0.000927 μmol /mg protein, the ergosterol level in our analysis was approximately 72-fold that of cholesterol. While the detection of cholesterol in yeast cells is normally attributed to laboratory contamination, it has also been validated that yeast strains can synthesize trace amounts of cholesterol when adenosylmethionine: Δ^{24} -sterol-C-methyl transferase is defective (Xu and Nes, 1988- PMID: 3046617).

For ion mode where the data obtained, indeed, these lipid classes were detected in different ion modes and quantitated in different injections. Kindly refer to our responses to the method description.

The lipids presented were chosen base on P-value less than 0.01 ($-\text{Log}_{10}$ p-value ≥ 2).

The Q1 and Q3 masses used to determine the shown lipid species and the retention times were provided in Supplementary Table S3.

Information of the checklist from the reference paper (doi:10.1038/s42255-022-00628-3) was addressed. Kindly refer to our responses to the method description.

2) “ - *The major effect on sphingolipid levels resulting from a CSG2 deletion is the depletion of MIPC and M(IP)2C with a concomitant increase in IPC levels. Complex sphingolipid should be included in the lipidomics analysis. CSG2 mutants show a two-fold increase in PHS. According to the authors this should be sufficient to block autophagy. What are PHS levels in the other mutants analyzed? Does the inhibition of the ceramide synthase with fumoinisin B1 block autophagy?*”

Response: We thank the reviewer for these suggestions and comments. First, in our original analyses of the yeast lipidome, complex sphingolipids were not included because of their relatively lower endogenous abundances compared

to PhytoCers, which could not accommodate these lipids in a single injection at a specified sample concentration. In our revision, we made an additional injection at higher sample concentration to analyze complex sphingolipids such as MIPC. Results showed that Csg2 deletion caused decrease of MIPC (Fig. S5g). In *csg2Δ* cells, autophagy was blocked; in *csg2Δ tsc10^{ts}* cells, autophagy was restored, indicating that accumulation but not decrease of certain sphingolipid(s) is the reason for autophagy blockage caused by Csg2 deletion. In *csg2Δ lip1^{ts}* or *csg2Δ aur1^{ts}* cells, autophagy was not restored, indicating that it is the accumulation of PHS or DHS (or their phosphorylated forms) but not PHCer/DHCer or downstream complex sphingolipids that cause autophagy blockage. Further deletion of Sur2 but not Lcb4/5, Ssc7 or Ccc2 can restore autophagy in *csg2Δ* cells, indicating that it is the accumulation of PHS that caused autophagy blockage in *csg2Δ* cells.

Second, we analyzed the levels of PHS in mutant cells with *tsc10^{ts}*, *lip1^{ts}*, and *aur1^{ts}* double mutation with *csg2Δ*. Results showed that *tsc10^{ts}* double mutation with *csg2Δ* caused decrease of PHS compared with that in *csg2Δ* cells, while *lip1^{ts}* or *aur1^{ts}* double mutation with *csg2Δ* caused increase of PHS (Fig. S6g). This is in line with the positions of Tsc10, Lip1 and Aur1 in the process of sphingolipid synthesis (Fig. 6c), with Tsc10 at the upstream of PHS for its synthesis and Lip1 and Aur1 at the downstream of PHS for its metabolism. The low levels of PHS in *tsc10^{ts} csg2Δ* cells and high levels of PHS in *lip1^{ts} csg2Δ* or *aur1^{ts} csg2Δ* cells are in line with the results that autophagy in *tsc10^{ts} csg2Δ* cells was restored but autophagy in *lip1^{ts} csg2Δ* or *aur1^{ts} csg2Δ* cells was not restored (Fig. 6e).

Third, we analyzed whether autophagy is blocked by ceramide synthase inhibitor Fumonisin B1. Results showed that treatment of high amount of Fumonisin B1 caused blockage of autophagy (Fig. S6c), which was in line with the results that autophagy was blocked in ceramide synthase *lip1* mutant cells (Fig. 6b).

3) “ - *The authors speculate in the discussion that increased PHS levels inhibit the TORC1 complex to block the formation of autophagosomes. This would be an important point to strengthen the manuscript. Assays to determine TORC1 activity in yeast are available and this should be tested. How do the authors explain the accumulation of PHS in the ER to affect the vacuolar TORC1 complex?*”

Response: We thank the reviewer for this suggestion. We analyzed the TORC1 activity in *csg2Δ* cells (PHS increased) by detecting the activation (phosphorylation) of its downstream protein Sch9 (ortholog of mammalian S6 kinase). Results showed that phosphorylation of Sch9 was increased in *csg2Δ* cells under starvation conditions and this increase was abolished by further deletion of Sur2 (Fig. S7u). This was in line with results showed that under starvation conditions PHS was increased in *csg2Δ* cells and this increase was abolished by further deletion of Sur2 (Fig. S6k). These results suggested that TORC1 activity is activated by increased PHS in *csg2Δ* cells, and as known TORC1 activation inhibits autophagy.

For the potential mechanism(s) of accumulation of PHS in ER affects the vacuolar TORC1 activity, we speculated that the increased PHS may traffic out of ER to cytosol, leading to PHS encounters TORC1 or its upstream regulators, which then activates TORC1 activity. Another possibility is that PHS indirectly inhibits TORC1. The increased PHS in ER disturbs the capability of ER, leading to a certain status of ER (such status disrupts ER-vacuole contact?), which is a signal for activation of TORC1. In the future, it will be interesting to further confirm and investigate how PHS activates TORC1.

4) “ - *Previous manuscripts show the effect of CSG2 deletion on stability of the MIPC synthase. This is completely ignored in the manuscript and especially the discussion.*”

Response: We thank the reviewer for this suggestion. The protein levels of MIPC synthase, Csg1 (also named Sur1) and Csh1, were analyzed in *csg2Δ* cells after starvation. Results showed that the protein levels of Csg1 (also named Sur1) and Csh1 were not affected by deletion of Csg2 (Fig. S7m, n), in contrast with the specific disruption of Aur1 (Fig. 7b). Although double deletion

of Csg1 and Csh1 caused blockage of autophagy (Fig. S6d) and the increase of PHS (Fig. S6e), the protein levels or the cellular distribution of Aur1 were not affected by Csg1/Csh1 deletion (Fig. S7i, j). We speculated that although both Csg2 deletion and Csg1/Csh1 deletion caused blockage of autophagy, the behind mechanisms are different. Deletion of Csg2, the ER calcium channel, induced the accumulation of calcium in ER, leading to disruption of Aur1 and followed increase of PHS that caused blockage of autophagy. Deletion of Csg1/Csh1, the MIPC synthases in sphingolipid synthesis pathway, induced the increase of PHS and followed blockage of autophagy.

Minor points:

1) “ - According to the figure legends, temperature sensitive mutants were grown at non-permissive temperature for 16 hours. How many cells are still alive after such a long exposure to the non-permissive temperature?”

Response: We thank the reviewer for this suggestion. We analyzed the cell viabilities of these temperature sensitive mutants after grown at non-permissive temperature. These temperature sensitive mutants were spotted to YPD plates at permissive temperature (24°C) before and after 16h of growth at non-permissive temperature (37°C). Results showed that these temperature sensitive mutants showed similar growth rate as WT cells even after 16h of growth at non-permissive temperature (Fig. S6f), indicating that the autophagy defect in these mutant cells was not caused by cell death. This is in line with the results that autophagy was normal in Tsc10 mutant cells and was blocked in Lip1 or Aur1 mutant cells (Fig. 6b). IF these temperature sensitive mutants were dead after treatment, they should exhibit autophagy blockage unanimously.

2) “ - The authors should be careful regarding their lipid annotation. For example, in figure 7a in the upper panel, PHS is used. In the other panels the authors use “sphingosine”. Sphingosine has a double bond and does not exist as a molecule in yeast cells. Only PHS and DHS are present and according to the results of the SUR2 mutants it makes a big difference for the outcome. Thus, the nomenclature should be treated very carefully.”

Response: We thank the reviewer for this suggestion. We are sorry for this mistake of lipid annotations. The “sphingosine” was not used when PHS or DHS in yeast was indicated. Instead, the annotations of PHS and DHS were used when trying to indicate them in yeast cells.

3) “ - *The model in figure 1a is also inaccurate. Why does the ER has such a big lumen. I assume that is supposed to be the nucleus. This should be visualized with more care.*”

Response: We are sorry for this mistake of ER illustration in model scheme in Fig. 1a. Indeed, the lumen should be nucleus. This has been corrected in schemes in Fig. 1a and Fig. 4a.

4) “ - *Can the others include a control that shows the functionality of GFP tagged Csg2, eg. in autophagy (pho8D60 assay) or starvation resistance?*”

Response: We thank the reviewer for this suggestion. GFP tagged Csg2 (Csg2-GFP) was expressed in *csg2Δ* cells and autophagy activity (investigated by pho8D60 assays) and starvation resistance were analyzed. Results showed that Csg2-GFP restored autophagy activity and starvation resistance in *csg2Δ* cells (Fig. S2a, b), indicating that the GFP tagged Csg2 was functional.

REVIEWER COMMENTS

Reviewer #1 (Remarks to the Author):

The manuscript by Liu et al. presents a detailed analysis on the calcium-controlled mechanisms regulating autophagy in yeast cells. They characterize the calcium channel Csg2 and show its localization in the ER membrane and its specificity for calcium ions. Depletion of Csg2 (Δ csg2 yeast cells) results in a significant increase in calcium levels in the ER. The authors show that depletion of Csg2 inhibits autophagy in starved yeast cells. By using a variety of different experimental approaches, the authors show that the blockage of autophagy is based on the accumulation of $[Ca^{2+}]_{ER}$. Lipidomics analysis revealed elevated levels of phytosphingosine and ceramides and a drastic decrease in MIPC in starved Δ csg2 cells. By deleting Sur2 (which is upstream of PHS synthesis) they experimentally downregulate the endogenous elevated PHS levels in Δ csg2 cells and show that in these Δ csg2/ Δ sur2 cells autophagy is reconstituted. While re-expression of Sur2 in Δ csg2/ Δ sur2 rescued the block of autophagy. Further characterization of the Δ csg2 cells at protein level point to destabilization of Aur1, which is located downstream of PHS and catalyzes the synthesis of complex phytoceramids. The authors further analyse the fate of Aur1 protein in starved Δ csg2 cells and show its mislocalization and subsequent proteasomal degradation. According to the data the authors sum up their results presenting a thesis according to which the deletion of csg1 results in elevated levels of $[Ca^{2+}]_{ER}$, which lead to destabilization and degradation of Aur1, resulting in elevated PHS levels, which eventually block autophagy, possibly via activation of TORC1.

All the data provided are based on different experimental approaches and are described and shown in detail. All the experiments are well designed and are clearly presented in this revised manuscript.

correction required

in line 466: instead of "The interaction between Aur1 and Csg2..." it should be "The interaction between Aur1 and Kei1...."

in line 465: "kei1" should be "Kei1"

Reviewer #2 (Remarks to the Author):

The authors have addressed all of my concerns.

Reviewer #3 (Remarks to the Author):

The authors have adequately addressed my concerns, except with respect to the localization of Aur1 and Kei1. If the authors can handle the following points, I think that the paper is now suitable for publication in the Nature Communications.

Previous detailed studies of Aur1 and Kei1 (Levine et al., 2000, DOI: 10.1091/mbc.11.7.2267; Lisman et al., 2004, DOI: 10.1074/jbc.M306119200; Sato et al., 2009, DOI: 10.1091/mbc.e09-03-0235, and also in database of SGD [<https://yeastgfp.yeastgenome.org/getOrf.php?orf=YKL004W>]) have shown that Aur1 localizes to the Golgi in exponentially grown cells in nutrient-rich media. Moreover, it has been shown that Aur1 does not return to the ER when ER-to-Golgi transport is blocked (Levine et al., 2000, DOI: 10.1091/mbc.11.7.2267). Recent studies have also shown that Aur1 localizes to the Golgi (Liu et al., 2017, DOI: 10.1083/jcb.201606059), and I think that the authors will not be able to reach a consensus to argue that Aur1 localizes to non-Golgi organelles such as ER unless they can show certain conditions for its localization to the ER.

The authors showed that in nutrient rich conditions (SD-N 0h), both Aur1 and Kei1 exhibited merge distribution with ER marker (Fig. S7d, e), and speculate that Aur1 and Kei1 may localize both at ER and/or Golgi, and in different yeast background or culture situations, their localizations vary.

The localization of enzymes that catalyze lipid synthesis is crucial to understanding the regulation of lipid metabolism, homeostasis, and the function of lipids. The yeast strains used by the authors in this study is the commonly used experimental yeast BY strain, and the same BY strain has been used in a study showing localization of Aur1-GFP to the Golgi apparatus (Huh et al., 2003, DOI: 10.1038/nature02026). Therefore, it cannot be ruled out that the AUR1-GFP constructed by the authors may be dysfunctional. For the authors to claim that Aur1 is localized to the ER, they should provide experimental evidence under what culture conditions it becomes localized to the ER rather than the Golgi apparatus. Even in nutrient rich media, the localization may differ if the yeast is in different growth phases (lag phase, diauxic phase, stationary phase). It has been reported that the localization of Isc1, an enzyme involved in the catabolism of complex sphingolipids is regulated in a growth-dependent manner (Vaena de Avalos et al., DOI: 10.1074/jbc.M309586200). The cause of the difference in localization should be experimentally demonstrated.

Reviewer #4 (Remarks to the Author):

I would like to express my gratitude to the authors for carefully addressing my concerns. They have significantly improved the annotation of their lipidomic analysis and provided more information, making the manuscript more comprehensible.

However, my main concern still remains that the functional link between calcium accumulation, autophagy, and the role of phytosphingosine in this process remains unclear. I am still missing information on how the accumulation of PHS blocks autophagy. Although the link to TORC1 activation has been strengthened, the overall mechanism remains rather descriptive.

Nevertheless, I believe that this study could pave the way for new investigations into the connection between autophagy and sphingolipid metabolism and could, therefore, be published in Nature Communications

Point-by-point responses to the reviewers

All the authors thank the appreciation of our revision work from the reviewers. The insightful comments and suggestions from the reviewers will help us to promote the findings on calcium, sphingolipids (PHS) and autophagy regulation in the future. The text correction (“Csg2” to “Aur1” and “kei1” to “Kei1”) mentioned by Reviewer #1 has been done (line 469, 470). The remaining point of Reviewer #3 was addressed, as shown below.

Reviewer #3

We thank the reviewer for the time and positive comments on our revised manuscript.

1) The remaining point is on the cellular localization of Aur1, “.....*For the authors to claim that Aur1 is localized to the ER, they should provide experimental evidence under what culture conditions it becomes localized to the ER rather than the Golgi apparatus. Even in nutrient rich media, the localization may differ if the yeast is in different growth phases (lag phase, diauxic phase, stationary phase). It has been reported that the localization of Isc1, an enzyme involved in the catabolism of complex sphingolipids is regulated in a growth-dependent manner (Vaena de Avalos et al., DOI: 10.1074/jbc.M309586200). The cause of the difference in localization should be experimentally demonstrated.*”.

Response: We also noticed that the ER-localization of Aur1 in cells under the culture conditions we used was different from reported studies. Actually, this was why we tested both the N-terminal GFP tagged Aur1 and the C-terminal GFP tagged Aur1 for localization detection, which both showed ER distribution (Fig. 7c and S7c). Since Aur1-GFP can be co-immunoprecipitated with Kei1 in WT cells (Fig. 7e), we speculated that Aur1-GFP is correctly folded and thus functional.

We then experimentally demonstrated under what kinds of conditions Aur1 showed ER or Golgi localization.

First, we detected the localization of Aur1-GFP in different background of budding yeast (DF5, BY4742, W303), results showed that the localization of Aur1-GFP was the same as that in BY4741 cells (Revision Fig. 1, see below).

Second, we detected the localization of Aur1-GFP in different culture temperatures and culture media (YPD or synthetic medium), results showed that Aur1-GFP still showed ER distribution (Revision Fig. 2, see below).

Third, we detected the localization of Aur1-GFP in different culture times/growth phases. Yeast cells at log phase ($OD_{600}=0.8-1$) and cells before or after log phase were detected. Results showed that shortly after log phase, cells showed dots rather than ER distribution of Aur1-GFP (Revision Fig. 3, see below). We then detected the co-localization of Aur1-GFP and Golgi marker Vrg4-RFP in cells at stationary phase (16 hour after log phase), and results showed that Aur1-GFP showed co-localization with Golgi Vrg4-RFP in cells at stationary phase (Revision Fig. 4, see below or Fig. S7d).

Therefore, we speculated that the localization of Aur1 differs in yeast cells in different growth phases (we highly thank the reviewer for this suggestion). It has been reported that the localization of *Isc1*, an enzyme involved in the catabolism of complex sphingolipids is regulated in a growth-dependent manner, predominantly in the ER during early growth but associated with mitochondria in late logarithmic growth (DOI: 10.1074/jbc.M309586200) (thank the reviewer for pointing this reference). It is interesting that these enzymes in sphingolipid metabolism pathway change their localizations in different growth phases. We speculated that the changes of localization of these enzymes may be caused by certain lipids or modification associated with growth phases, and their activities may be changed after localization to different organelles.

We described the localization of Aur1-GFP in cells in stationary phase in the revised manuscript (Fig. S7d and line 458-463): “Aur1 showed ER localization in yeast cells in log phase (Fig. 7c and Fig. S7c), while showed Golgi localization in cells in stationary phase (Fig. S7d) as shown before¹²⁵⁻¹²⁹. Interestingly, localization of *Isc1*, an enzyme involved in sphingolipid catabolism that hydrolyzes complex sphingolipids to produce ceramide, is also regulated in a growth-dependent manner¹³⁰.”.

Revision Fig. 1

Revision Fig. 2

Revision Fig. 3

Revision Fig. 4

REVIEWER COMMENTS

Reviewer #3 (Remarks to the Author):

Thank you for your experiments in response to my concern. Since previous studies have shown that Aur1 is localized to the Golgi apparatus in cells growing exponentially in nutrient-rich medium, I had expected that the localization of Aur1 to the endoplasmic reticulum (ER), which was observed by the authors, would be a specific phenomenon in the stationary phase. However, contrary to my expectation, the authors have shown in the revised version that Aur1 localizes to the ER in the log phase and then to the Golgi apparatus as the cell enters the stationary phase.

After de novo synthesized in the ER, Aur1 is believed to be transported to the Golgi via COPII vesicles. It is presumed that if abnormally large numbers of proteins are synthesized in the ER, they can cause a traffic jam, possibly resulting in the observed ER localization of Aur1-GFP/GFP-Aur1. In general, protein biosynthesis is increased in the log phase, while it is suppressed in the stationary phase due to nutrient depletion. Therefore, the ER localization of Aur1 in the log phase and the Golgi localization in the stationary phase, which was observed by the authors, may be due to an abnormally high expression of the Aur1 protein.

The author's results were very surprising to me because they are opposite to previous results showing that Aur1 localizes to the Golgi in the log phase. Thus, I have checked carefully the method used by the authors to analyze the localization and found that for the analysis of Aur1 localization, the authors used a strain in which cells were transfected with a plasmid carrying a promoter of ADH1, which is used in constitutive overexpression systems. Therefore, it is possible that the results observed by the authors are the result of overexpression of Aur1 and are artifacts. The observed localization of Kei1 to the ER could be due to the same reason. This concern can be addressed by the following experiment.

- 1) Perform the experiment with a plasmid (CEN) that expresses AUR1-GFP/GFP-AUR1 under its own promoter instead of the ADH1 promoter.
- 2) The plasmid-transfected cells will have two copies together with the endogenous AUR1, so it would be much better to do the experiments with a strain that has GFP inserted into the AUR1 of the genome.
- 3) Alternatively, the experiment should be performed in cells that have been treated with cycloheximide, which inhibits protein synthesis, and cultured for several hours to allow proteins to reach their native location.

If the authors would show that Aur1-GFP/GFP-AUR1 is localized in the Golgi by either of these methods, it is most likely that Aur1 localization to the ER is an artifact, a phenomenon that occurs specifically

when overexpressed under the promoter of ADH1, and does not reflect Aur1 localization in normal cells. Therefore, I would recommend reconfirmation by one of the above methods, and if the authors found that the ER localization of Aur1 is due to overexpression, the manuscript should be revised to reflect that result. Alternatively, if the authors do not address this issue, I recommend removing all data showing localization of Aur1 and Kei1 shown in Fig. 7 and Supplemental Fig. 7, and correcting the manuscript including Discussion and the model in Fig. 7.

Point-by-point responses to the reviewers

Reviewer #3

We thank the insightful comments and suggestions from the reviewer, which inspire the directions of our research in the future.

1) The remaining point is on the cellular localization of Aur1, especially on the potential reasons for the ER and Golgi localization of Aur1.

Response: We highly thank the reviewer for the careful and serious examinations on the localization of Aur1. Indeed, this inconsistency of Aur1 localization we observed had been a big puzzle to us. At first, we ruled out the possibility that the N-terminal or C-terminal GFP-tagging influences the localization of Aur1. Then different genetic background of budding yeast, culture media or temperatures were also ruled out. The Golgi localization of Aur1 in stationary cells was surprising to us too. We observed this at several time points after cells enter log phase (3h, 16h, and 24h after log phase) and the Golgi localization of Aur1 was detected. Thank the reviewer for proposing that the protein biosynthesis is suppressed in the stationary phase, we realize that the possible reason behind may be the low protein levels of Aur1 in stationary phase cells. We confirmed that the Aur1 protein in stationary phase cells was at very low levels (Revision.Fig.1 a). As suggested, we generated a strain that has GFP C-terminally inserted into the AUR1 of the genome. Results showed that Aur1-GFP^{Chromosomal} showed low protein levels and exhibited Golgi localization (Revision.Fig.1 b, c).

We highly thank the comments and suggestions from the reviewer, which inspire the directions of our research in the future and make us think more thoughtfully and carefully, especially when we try to figure out how the calcium accumulation in ER causes disruption of sphingolipid metabolism. We will address the issue of Aur1 localization in the future with more time and efforts. We learned a lot from the scientific views of the reviewer, thanks.

As suggested, we removed data showing localization of Aur1 and Kei1 (and also other sphingolipid synthases) shown in Fig. 7 and Supplemental Fig. 7, and corrected the manuscript including Discussion and the model in Fig. 7.